# INSIGHTBENCH: EVALUATING BUSINESS ANALYTICS AGENTS THROUGH MULTI-STEP INSIGHT GENERATION

**Gaurav Sahu**[1,6] **Abhay Puri**[1] **Juan Rodriguez**[1,2,4] **Amirhossein Abaskohi**[1,5]

**Mohammad Chegini**[7] **Alexandre Drouin**[1] **Perouz Taslakian**[1]

**Valentina Zantedeschi**[1] **Alexandre Lacoste**[1] **David Vazquez**[1] **Nicolas Chapados**[1]

**Christopher Pal**[1,2,3] **Sai Rajeswar Mudumba**[1,2] **Issam Hadj Laradji**[1,5]

[1]ServiceNow Research
[2]Mila - Quebec AI Institute
[3]Canada CIFAR AI Chair
[4]École de Technologie Supérieure
[5]University of British Columbia
[6]University of Waterloo
[7] University of Victoria

## ABSTRACT

Data analytics is essential for extracting valuable insights from data that can assist organizations in making effective decisions. We introduce *InsightBench*, a benchmark dataset with three key features. First, it consists of 100 datasets representing diverse business use cases such as finance and incident management, each accompanied by a carefully curated set of insights planted in the datasets. Second, unlike existing benchmarks focusing on answering single queries, *InsightBench* evaluates agents based on their ability to perform end-to-end data analytics, including formulating questions, interpreting answers, and generating a summary of insights and actionable steps. Third, we conducted comprehensive quality assurance to ensure that each dataset in the benchmark had clear goals and included relevant and meaningful questions and analysis. Furthermore, we implement a two-way evaluation mechanism using LLaMA-3 as an effective, open-source evaluator to assess agents' ability to extract insights. We also propose *AgentPoirot*, our baseline data analysis agent capable of performing end-to-end data analytics. Our evaluation on *InsightBench* shows that *AgentPoirot* outperforms existing approaches (such as Pandas Agent) that focus on resolving single queries. We also compare the performance of open- and closed-source LLMs and various evaluation strategies. Overall, this benchmark serves as a testbed to motivate further development in comprehensive automated data analytics and can be accessed here: `https://github.com/ServiceNow/insight-bench`.

## 1 INTRODUCTION

Businesses frequently leverage vast datasets to perform data analytics to uncover insights, discover patterns, and analyze trends in order to support effective decision-making (McAfee & Brynjolfsson, 2012; Colson, 2019; Bean, 2022). The task of end-to-end data analysis starts with stating a high-level goal; the analyst then alternates between identifying key questions to explore, and extracting valuable insights from their answers, working towards the goal. This iterative process continues until they have a comprehensive summary of insights and recommended actions (shown in Figure 1).

Analysts often carry out such tasks by leveraging tools like Jupyter notebooks and dashboards (Bean, 2022; Yin et al., 2023), which, unfortunately, require both significant human effort as well as data science and domain expertise. Fortunately, agents based on large language models (LLMs) have emerged as promising

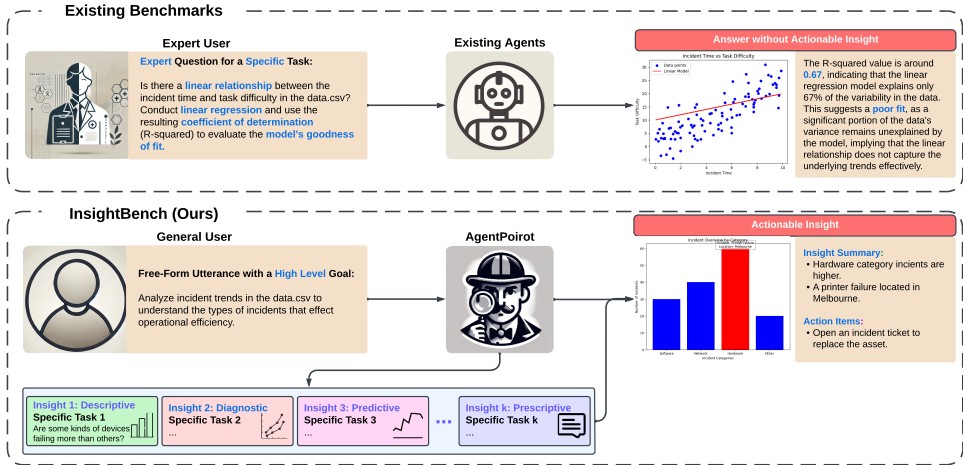

**Figure 1:** Existing benchmarks (top) assess the agents' ability to solve a highly specific data analytics query with pre-defined output templates. They often require an expert user to ask the questions. InsightBench (bottom), on the other hand, evaluates the LLM-based agents on the complete comprehensive data analytics processes. This includes evaluating the agents' ability to answer high-level questions by a general user, recommend the best specific tasks to address a specific goal, extracting insights across descriptive, diagnostic, predictive, and prescriptive categories, and summarizing both findings and recommending next steps (as demonstrated by AgentPoirot).

assistants to users in performing data science tasks (OpenAI, 2022; 2023). But these agents (such as Pandas Agent (LangChain, 2024) and Code Interpreter (OpenAI, 2024)) only focus on solving narrow, single-step code-completion (or data analytics) tasks (E.g., "What is the $R^2$ value for a linear regression model on this dataset?").

One of the reasons most existing agents have limited data analysis capabilities is because current benchmarks focus on evaluating agents on single-step code-completion tasks (Lai et al., 2022; Chen et al., 2021a; Hu et al., 2024). Another reason is that it is challenging to evaluate LLM-based agents accurately without including substantial information about the problem details, expected code structure, and expected outputs (such as identifying variable correlation) in the prompt. Recent works such as InsightPilot (Ma et al., 2023) and InsightLens (Weng et al., 2024) propose agents that can perform end-to-end data analytics, but they were evaluated using human assessments, which require significant effort and often lack consistency due to variability in individual biases and interpretations of evaluation criteria.

Constructing a standardized, automated benchmark to address the consistency and reliability concerns involves tackling two main challenges. The first concern is to ensure that the agent explores the most informative and interesting questions, given the data and goals. Second, we must ensure that the insights derived from these questions are accurate and relevant. These challenges stem from the fact that there are many ways to ask interesting questions about the data and different ways to interpret their answers. To address these issues, we need datasets with clearly defined data and goals, equivalent to analysis performed by an expert data scientist.

This inspired us to propose *InsightBench*, which consists of 100 datasets whose structures have been acquired from the ServiceNow (ServiceNow, 2024) platform, which focuses on business operations and workflows. This platform allows us to generate synthetic tabular data that mimics real-life enterprise data. We used it to create datasets for Incident, User, Finance, Inventory, and Enterprise Goal Management. We chose this data because an agent capable of effectively extracting insights from it can be deployed to help organizations optimize their business operations.

To ensure the quality of goals and questions in our datasets, we had expert annotators set clear goals and manipulate the data to plant interesting insights. Each insight includes a question, generated code, and plot values from which the insight is extracted. The annotators then aggregate these insights, summarize them, and provide recommended actions. We ensure that our insights are discoverable, so a good LLM-based agent performing analytics on the data should recommend similar questions and extract similar insights as the ones planted by the annotators.

To effectively score the agent's ability to predict the most relevant insights, we need a scoring method that can compute the semantic and factual similarities between two free-form texts. In our case, we need to score how closely the predicted insights match the ground-truth insights.

Therefore, we draw inspiration from G-Eval which is a state-of-the-art technique that uses GPT-4 to evaluate the quality of generated texts in a manner that aligns closely with human judgment (Liu et al., 2023). Since GPT-4 is costly and closed-source, we replace it with Llama-3 (Touvron et al., 2023) and call the technique LLaMA-3-Eval. Our experiments show that LLaMA-3-Eval and G-Evalproduce consistent outcomes when ranking the performance of agents in extracting insights.

We benchmark LLM-based agents on *InsightBench*, where we run a variant of the Pandas Agent that can perform end-to-end data analysis by recommending questions and extracting insights from their answers. We also propose a baseline agent, *AgentPoirot*, which can extract deeper insights and questions by carefully designed prompts covering Descriptive (what happened), Diagnostic (why it happened), Predictive (what will likely happen), and Prescriptive (what actions to take) analytics. Our evaluation shows that *AgentPoirot* obtains a higher LLaMA-3-Eval score compared to Pandas Agent.

To summarize, our paper makes the following contributions:

- We propose *InsightBench*, the first benchmark specifically designed to evaluate LLM-based agents for performing end-to-end, multi-step data analytics.
- We present a comprehensive set of experiments contrasting open-source and closed-source models and different prompting strategies for evaluation on our benchmark. Our results show that *AgentPoirot* outperforms existing analytics methods such as Pandas Agent and LLaMA-3-Eval is a feasible alternative to using the closed-source G-Eval.

## 2   *InsightBench* – AN ENTERPRISE DATA ANALYSIS BENCHMARK

*InsightBench*'s data were derived from the ServiceNow (ServiceNow, 2024) platform, which is used to manage business workflows crucial for enterprise operations. It consists of tables that store and manage records and data entities relevant to various organizational functions. Common tables in ServiceNow include incident tables (for incident records), user tables (for user information), and asset management tables (to oversee infrastructure). *InsightBench* consists of 100 tabular datasets acquired from the ServiceNow platform covering five distinct themes (see Figure 3).

**Overview of Benchmark Creation.**   *InsightBench* is an automated data analysis benchmark that can rigorously assess the performance of LLM-based data analysis agents in real-world scenarios. Building *InsightBench* consisted of four stages: 1) selecting relevant data tables from the ServiceNow data tables and extracting a list of relevant columns to define the schema (Section 2.1.1), 2) formulating trends to inject in the data (Section 2.1.2), 3) creating synthetic data entries to populate the tables that follow the trend formulated in the previous step (Section 2.1.3), and 4) creating a ground-truth analysis notebook for the generated data (Section 2.1.4). See Figure 2 for an overview of the multi-stage process. A detailed list of the datasets sorted by themes is presented in Appendix B.3. We propose a multi-step evaluation mechanism to measure the performance of an agents on *InsightBench* (See Section 2.2)

### 2.1   CREATING *InsightBench* DATASETS

Each dataset consists of 500 synthetically generated entries, stored as a CSV file. We choose a size of 500 entries to keep the volume of data manageable for analysis and substantial enough to simulate typical enterprise data loads. To emulate a realistic enterprise environment, *InsightBench* uses a hybrid approach combining actual data structures with synthetic insights.

### 2.1.1   DATA TABLES

To create the datasets, we first select relevant data tables from ServiceNow system tables. As a guiding example, consider an incident table, which contains records of any disruption to normal service operations. The incidents can range from server outages to hardware malfunctions, such as a non-operational printer. This table is structured with fields (aka columns) relevant to managing the lifecycle of an incident, such as the time it was opened, the description of the incident, and assigned agents (See Figure 2(a)). We

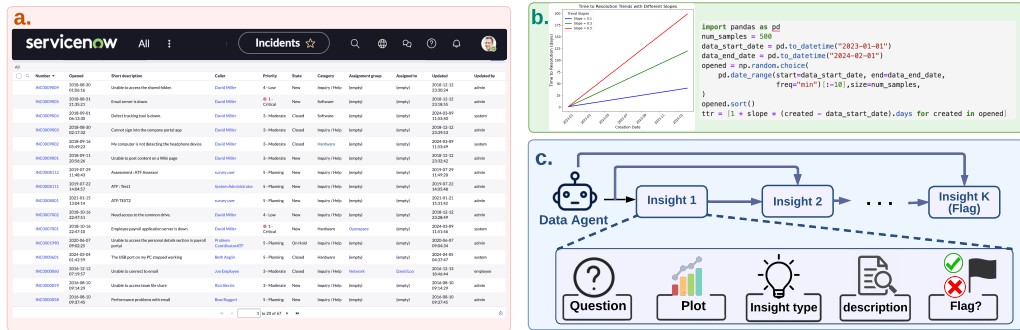

**Figure 2: Stages of Benchmark Creation: (a)** Shows a demo incidents table from ServiceNow for the Incident Management theme, detailing schema and data fields. **(b)** Demonstrates the process of creating one of the datasets in the benchmark that embeds a linear increasing trend in incident resolution times, also highlighting the role of the slope parameter in dictating the trend's strength. **(c)** Displays the multi-step annotation analysis, with each step involving a question, corresponding plot, insightful description, and classification of the insight type.

choose appropriate columns that allow us to plant realistic trends (as explained in the next subsection) while excluding irrelevant columns.

### 2.1.2 PLANTING INSIGHTS

We embed systematic anomalies and trends into the datasets to certain types of columns in the datasets, referred to as *controllable columns*. These trends were implemented using standard mathematical models or algorithms, such as regression, to ensure consistency and detectability of the planted insights. For instance, we manipulated the time-to-resolution (TTR) trend for incidents in a service management dataset to reflect an increasing trend over time. This was achieved by using a linear model to generate the TTR based on the creation date of each incident: $\text{TTR} = 1 + \text{slope} \cdot (\text{incident\_open\_date} - \text{data\_start\_date})$, where (slope) is a parameter that controls the rate of increase in resolution time (See Figure 2(b)).

**Types of Insights.** Every insight in *InsightBench* can be categorized into four types, where each type serves a specific function in data analysis, contributing uniquely to the utility of the insights derived.
1. **Descriptive**: These insights describe what has happened and are vital in summarizing large datasets into understandable plots. E.g., a plot of the distribution of incident categories over the past year.
2. **Diagnostic**: These insights explain the why or the cause behind observed trends using tools such as segmentation and correlation. E.g., a wordcloud of the most common words in incident description.
3. **Predictive**: These insights use statistical methods to forecast future outcomes based on past data. E.g., a plot forecasting the future volume of incidents based on current trends in resolution times.
4. **Prescriptive**: These types of insights suggest actions to tackle the current issues. E.g., an insight recommending strategies to mitigate this projected increase in incident volume.

### 2.1.3 SYNTHETIC DATA GENERATION

We employed two primary methods to populate the *non-controllable columns*–columns that are not directly manipulated to embed synthetic insights, where each method is chosen to maintain authenticity and alignment with real-world data characteristics. The entire data generation pipeline is implemented in Python, allowing for reproducibility and scalability. We now describe the two methods below:

- **Random Sampling:** For fields like IDs, categories of incidents or assets, transaction dates, and status codes, we used random sampling from carefully curated lists of plausible values.
- **Context Generation Using Large Language Models (LLMs):** To introduce complexity and realism, certain text-based fields like incident or goal descriptions and user feedback were generated using LLMs such as GPT-4 (OpenAI, 2023). These models were tasked with creating coherent and contextually relevant entries that align with the data schema, enhancing the datasets with natural-looking and appropriate text data.

We include more details in Appendix B.1, and the prompt structure used can be found in Appendix D.

### 2.1.4 GROUND-TRUTH ANALYSIS NOTEBOOKS

A key component of constructing *InsightBench* was developing 100 expert-annotated Jupyter notebooks, each tailored for a specific context, such as incident management or financial operations. The notebook structure and contents are as follows (see Figure 2(c)):

- **Dataset overview and a SMART Goal**: Each notebook begins with a comprehensive dataset overview, outlining its relevance and structure. It is accompanied by a SMART (Specific, Measurable, Attainable, Relevant, and Timely) goal (E.g., "Analyze the discrepancy and imbalance in the distribution of incidents assigned across categories," aimed at identifying the primary causes of hardware failures in an organization.)
- **Sequential Analysis**: The notebook contains a series of questions designed to uncover layers of insights gradually. Each question has a Python code block that generates a plot to answer the question, and each plot's data is summarized as JSON metadata outlining the insights. We ensure that each question builds sequentially onto the previous one, leading up to the final insight.
- **Insight Summary**: The final section of each notebook summarizes the findings and proposes actionable steps. This might include recommendations like "Open a Ticket" for further investigation.

### 2.2 EVALUATING AGENTS ON *InsightBench*

To evaluate agents on *InsightBench*, we compare the ground-truth annotations ($GT$) in Jupyter notebooks against the list of insights ($A$) provided by an agent. Formally speaking, we compare two natural language texts. While multiple metrics have been proposed in the past, like ROUGE (Lin, 2004), METEOR (Banerjee & Lavie, 2005), and BERTSCore (Zhang et al., 2019), they do not align well with human preferences. Recently, G-Eval (Liu et al., 2023) was shown to be highly aligned with human preferences for a variety of tasks, including text summarization. Since our task involves comparing two free-form sentences, we adopt an LLM-based evaluator as well. Specifically, we use LLaMA-3-Eval, a variant of G-Eval that uses `LLaMA-3-70b` instead of a GPT model. We use LLaMA-3-Eval as the primary metric to measure the correctness of agent-provided insights. We chose LLaMA-3-Eval over G-Eval because `LLaMA-3-70b` is open-sourced, allowing us to a) avoid API costs for using GPT models from OpenAI and b) fix the model weights to obtain a stable and reliable evaluation metric, unlike G-Eval, whose output scores can change due to periodic updates to GPT endpoints.

*InsightBench* performs a two-way evaluation:
1. *Summary-level.* To obtain the summary-level score, we simply compute the LLaMA-3-Eval score between the agent-provided summary of insights, and the ground-truth summary.
2. *Insight-level.* First, we select the most appropriate insight $a \in A$ to evaluate against a given ground-truth insight ($gt \in GT$), and then average the scores for all the ground-truth insights. The insight-level score can be expressed as in Equation 1, where $|GT|$ is the number of ground-truth insights in a dataset, and $\mathcal{M}$ represents LLaMA-3-Eval evaluator:

$$score = \frac{\sum_{gt \in GT} \text{argmax}_{a \in A} \mathcal{M}(gt, a)}{|GT|} \tag{1}$$

We average the summary-level and insight-level scores for all 100 datasets in *InsightBench* to obtain a measure of the agent's performance. Additionally, we compute scores using ROUGE-1 for comparison.

### 2.3 DATASET STATISTICS

The reason is that it seems like *InsightBench* covers a distinct range of business analytics themes and varying difficulty levels, each chosen to reflect typical scenarios encountered in enterprise settings The benchmark includes 100 datasets with a total of 475 insights spread across five key topics. The distribution across the topic is shown in Figure 3, and the data tables used are outlined in Appendix B.2.

**Distribution by Dataset Category.** The benchmark has 20 datasets on Incident Management, 20 datasets on Asset Management, 15 datasets on User Management, 15 datasets on Finance Management, 10 datasets on Goal Management, 10 datasets on Asset & User Management combined, and 10 datasets on Finance & User Management.

**Distribution by Difficulty.** The datasets are assigned a difficulty level on a scale from 1 (easy) to 4 (hard).

Easy (Level 1-2) comprises 30 datasets that primarily involve direct data retrieval and basic analysis. Medium (Level 3) includes 36 datasets that may require the integration of multiple data sources or applying moderate data transformations. Hard (Level 4-5) consists of 34 datasets that require the calculation of multiple intermediate quantities or significant transformation of data variables. A detailed list of datasets with their respective topics and difficulty range is outlined in Appendix B.3

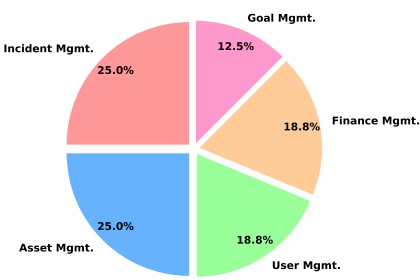

## 2.4 QUALITY ASSURANCE

To ensure our datasets are of high quality, we designed a questionnaire and an interface that displays our dataset to users. We had 20 volunteers with basic data science skills evaluate whether our datasets were accurately described, contained

**Figure 3:** *InsightBench* breakdown across five key thematic areas.

relevant and interesting questions, and provided substantial insights. The exact questions used, the interface design, and the results are shown in B.4. The average rating was 4 out of 5 across these questions, and we received helpful comments that improved parts of the dataset.

## 3 EXPERIMENTAL SETUP

The main experimental setup in our benchmarking process involves inputting the dataset schema and an overarching goal for each dataset in *InsightBench* and letting the agent perform exploratory data analysis. As discussed in Section 2, the goals are carefully designed to provide a meaningful signal to the agent without revealing the ground-truth answer.

In addition to the main experiments, we conduct the following ablation studies to understand the importance of different parameters of *InsightBench* and their effect on agent performance.

**Effect of Using Generic Goals.** Instead of using the carefully designed goal from the ground-truth annotations, we use the generic goal "I want to find interesting trends in this dataset." We aim to study the effectiveness of our goals in steering the agents toward performing meaningful data analysis.

**Effect of Trend Intensity.** We generate variations of Dataset 2 (see Table 2 for the complete list), which has the planted insight, "Time to resolution increases linearly over time." We vary the slope of the linear curve for that insight from -0.1 to 0.9. Through this experiment, we study whether an agent needs a trend to be blatantly obvious (high intensity) to retrieve or if the agent can retrieve nuanced trends (low intensity) as well. Notably, we also consider negative slopes in this experiment to test if the agents output a false positive (discovering a trend even when it is not present).

**Effect of Question Diversity.** We restrict the agent to generate only "descriptive" questions as follow-ups instead of letting it generate different types of questions.

**Effect of LLM Sampling Temperature.** We vary the sampling temperature of the LLM from 0 to 1.0 to study its effect on the agent's performance.

### 3.1 BASELINES

We run the following data analytics agents on InsightBench:

**a) Pandas Agent (PA)** (LangChain, 2024). A data science agent by LangChain optimized for question-answering [1]. Given a data frame and a question related to it, Pandas Agent first generates Python code and then executes it to produce the answer. In our experiments, we input the goal to the agent and let it iteratively generate question and their answers. We then pass the list of answers to the agent again and ask it to generate a summary of those answers.

**b) *AgentPoirot* (Ours).** We propose *AgentPoirot* that, given a dataset ($\mathcal{D}$) in *InsightBench* and a goal ($\mathcal{G}$), performs systematic data exploration aimed at achieving $\mathcal{G}$. Figure 7 shows the working of *AgentPoirot*:

---

[1]LangChain is an MIT Licensed Python library for building context-aware reasoning applications

1. **Extract Dataset Schema ($\mathcal{S}$):** The dataset schema contains information about every column in the dataset. *AgentPoirot* extracts the name, type, total unique values, and total `NA` values for every column. Additionally, if a column contains numerical data, it extracts the minimum, maximum, mean, and standard deviation; if a column contains dates, it extracts the minimum and maximum dates; otherwise, it extracts the top 5 unique values from the column.
2. **Generate Insights:** Given the triplet $(\mathcal{D},\mathcal{G},\mathcal{S})$, *AgentPoirot* uses the Question Generation Prompt to generate $k(=3)$ high-level questions. It answers each high-level question by first generating Python code and then interpreting the output of the code. Then, it generates $n(=4)$ follow-up questions for each high-level question and then answers them as well to obtain a total of generates $(n+1)\times k$ insights(see Figure 7 to see a complete overview of *AgentPoirot*).
3. **Generate Summary:** *AgentPoirot* summarizes the insights from the previous step (using Prompt 7.)

**c)** *AgentPoirot* **(Ours) w/ generic goal.** We use the generic goal, "I want to find interesting insights in this dataset," instead of the carefully designed goal and run *AgentPoirot* on *InsightBench* as in **b)**.

**d)** *AgentPoirot* **(Ours) non-diverse follow-ups.** We use Prompt 4 instead of Prompt 3 to generate only one type of follow-up questions and run *AgentPoirot* on *InsightBench* as in **b)**. Refer to Figure 7 in the Appendix for a visualization of our proposed method.

In addition to PandasAgent, we also considered the following baselines: Insightpilot (Ma et al., 2023), but neither the code nor the prompts are available; Data-Copilot (Zhang et al., 2023b), but it is limited to Chinese financial data, which isn't suitable for our task; OpenAgents (Xie et al., 2023), but the public version only allows 10 requests, which isn't enough for meaningful benchmarking; InfiAgent-Dabench (Hu et al., 2024), but we excluded it due to its extremely poor performance in this benchmark. It handles only simple tasks like "Generate Python code to compute the mean of a list," but struggles with more complex queries like "What is the distribution of incidents by category in this dataset?" Lastly, we considered using PowerBI [2], but it is a closed-source tool that is not suitable for large-scale benchmarking. We conducted a small-scale evaluation of PowerBI and found that it obtained low-quality insights and generated inconsistent outputs for the same prompt. Our benchmark emphasizes transparency and reproducibility, therefore we focussed on methods that the scientific community can easily inspect and scrutinize, which is not possible with the aforementioned baselines.

## 3.2 Implementation Details

We use Python's `openai` package to access the family of GPT models for our experiments and `vllm` to host `LLaMA-3-70b` [3] model on 4 A100 GPUs. We use different LLMs as backbones in our experiments to benchmark, including `gpt-4o`, `gpt-4-turbo`, `gpt-3.5-turbo`, and `llama-3-70b` [4]. All results are reported for the sampling temperature of 0.0, unless otherwise stated. We use the `evaluate` Python package to compute ROUGE-1 scores and Prompt 9 in Appendix D for computing LLaMA-3-Eval scores. We also fix the temperature of `LLaMA-3-70b` to 0 during evaluation. We repeat all our experiments for 5 seeds and report the mean and standard deviation in our results.

## 4 Benchmark Results

**Main Results.**  Table 1 shows the performance of different data analytics agents on *InsightBench*. First, we note that *AgentPoirot* outperforms Pandas Agent in terms of ROUGE-1 and LLaMA-3-Eval scores. We note that using `gpt-4o` consistently achieves the best overall performance. We further see that *AgentPoirot* with `LLaMA-3-70b` outperforms *AgentPoirot* with `gpt-3.5-turbo` and is close to `gpt-4-turbo` in terms of LLaMA-3-Eval scores.

Our performance analysis by dataset categories (Figure 4a) shows agents performing best in "Asset Management," "Goal Management," and "Finance Management." While *AgentPoirot* with `LLaMA-3-70b` underperforms on combined datasets due to context limitations, PandasAgent excels in these categories. Performance varies by difficulty (Figure 4b), with strong results on "Easy" and "Medium" tasks but declining on "Hard" ones. Across insight types (Figure 4c), agents show decreasing performance from

---

[2] `https://app.powerbi.com/home`
[3] `https://github.com/meta-llama/llama3`
[4] We use `gpt-4-turbo-2024-04-09` and `gpt-3.5-turbo-0125`, specifically.

**Table 1:** Performance of different agents on *InsightBench*. All results are for 5 different seeds.

| Agent | Insight-level Scores | | Summary-level Scores | |
|---|---|---|---|---|
| | ROUGE-1 | LLaMA-3-Eval | ROUGE-1 | LLaMA-3-Eval |
| PA (gpt-4o) | **0.35** $\pm0.03$ | 0.54 $\pm0.01$ | 0.35 $\pm0.01$ | 0.40 $\pm0.04$ |
| Ours (gpt-4o) | 0.32 $\pm0.02$ | **0.60** $\pm0.03$ | **0.37** $\pm0.09$ | **0.44** $\pm0.03$ |
| - w/ generic goal | 0.30 $\pm0.03$ | 0.40 $\pm0.03$ | 0.30 $\pm0.08$ | 0.33 $\pm0.12$ |
| Ours (gpt-4-turbo) | 0.30 $\pm0.02$ | 0.56 $\pm0.02$ | 0.35 $\pm0.08$ | 0.35 $\pm0.04$ |
| - w/ generic goal | 0.28 $\pm0.01$ | 0.38 $\pm0.03$ | 0.29 $\pm0.02$ | 0.27 $\pm0.11$ |
| Ours (gpt-3.5-turbo) | 0.34 $\pm0.01$ | 0.50 $\pm0.02$ | 0.27 $\pm0.14$ | 0.31 $\pm0.06$ |
| - w/ generic goal | 0.30 $\pm0.02$ | 0.36 $\pm0.03$ | 0.24 $\pm0.03$ | 0.25 $\pm0.06$ |
| Ours (llama-3-70b) | 0.33 $\pm0.02$ | 0.52 $\pm0.04$ | 0.36 $\pm0.01$ | 0.33 $\pm0.01$ |
| - non-diverse follow-ups | 0.29 $\pm0.01$ | 0.51 $\pm0.03$ | 0.32 $\pm0.01$ | 0.28 $\pm0.03$ |
| - w/ generic goal | 0.27 $\pm0.03$ | 0.35 $\pm0.01$ | 0.23 $\pm0.03$ | 0.23 $\pm0.02$ |

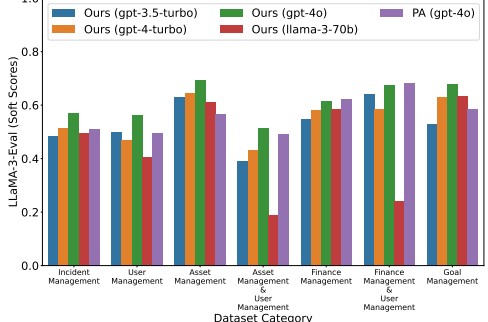

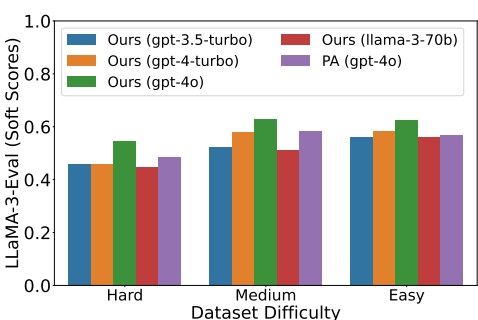

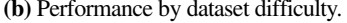

(a) Performance by dataset category.  (b) Performance by dataset difficulty.

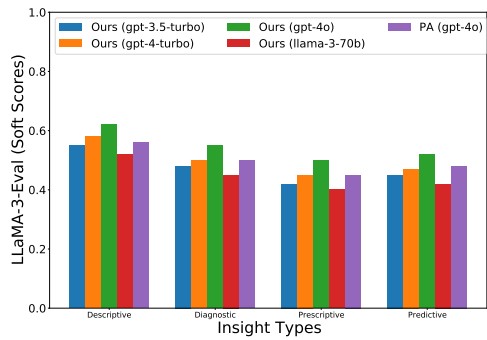

(c) Performance by insight type.

**Figure 4:** Performance of different agents on *InsightBench* grouped by difficulty and dataset category.

Descriptive (0.52-0.62) through Diagnostic, Prescriptive, and Predictive insights. Table 5 in Appendix C provides qualitative comparisons between *AgentPoirot* and PandasAgent.

**Effect of Using Generic Goals.** Table 1 also shows the performance of *AgentPoirot* that uses a generic goal. We notice a drastic overall decrease in the ROUGE-1 and LLaMA-3-Eval scores for all the backbones. For instance, the insight-level LLaMA-3-Eval score for *AgentPoirot* (`gpt-4o`) drops from 0.60 ($\pm$ 0.03) to 0.40 ($\pm$ 0.03) and the summary-level LLaMA-3-Eval score drops from 0.44 ($\pm$0.03) to 0.33 ($\pm$0.12) (see "Ours (gpt-4o)" v/s "Ours (gpt-4o) w/ generic goal" rows in Table 1). This highlights the importance of including a SMART goal in the agent's performance.

**Effect of Exploring Diverse Questions.** Table 1 includes the performance of *AgentPoirot* when generating only a single type of follow-up questions (see "Ours (llama-3-70b) non-diverse follow-ups"). We observe a notable decrease in ROUGE-1 scores and a slight decrease in LLaMA-3-Eval scores, confirming that discovering a diverse set of questions during data analysis leads to better performance.

**Effect of Trend Intensity** Figure 8b shows the performance of *AgentPoirot* with different backbones on variations of dataset 2 in *InsightBench*. First, we note that all the models fail to discover insight when the slope is less than 0.1 (including negative slopes). This suggests that our model did not generate false positives (we assume the flag is discovered if LLaMA-3-Eval is greater than 0.5. Overall, for slope greater than 0.1, `gpt-4o` and `gpt-4-turbo` discovered the insight every time, `LLaMA-3-70b` discovered it nearly half the times, and `gpt-3.5-turbo` had a particularly hard time discovering the insight. While both `LLaMA-3-70b` and `gpt-3.5-turbo` have difficulty discovering the trend sometimes (as showcased by the fluctuations in LLaMA-3-Eval scores), `LLaMA-3-70b` shows a better overall discovery rate of the insight compared to `gpt-3.5-turbo`.

**Effect of Sampling Temperature.** Figure 8a in Appendix C shows the performance of *AgentPoirot* with `LLaMA-3-70b` for different sampling temperatures. The performance of the agent peaks at temperature 0.2 before it starts to decrease. Higher temperatures also lead to unstable performance, as indicated by bigger standard deviations in the plot. Overall, we note that the temperature should be low but not completely 0 for optimal agent performance on *InsightBench*.

**G-Eval v/s LLaMA-3-Eval** Table 4 in Appendix C shows results for a small-scale study where we compare G-Eval and LLaMA-3-Eval. We note that both scores are similar in range and follow the same trend for all agents. This suggests that LLaMA-3-70b is a good alternative to using gpt-4o for evaluation.

**One-to-Many v/s Many-to-Many Evaluation.** We conduct another study to compare our approach with an alternative many-to-many version, where, instead of computing a score for each ground truth and prediction pair, we first prompt the LLM to match every ground truth response with an appropriate prediction. We perform this experiment because the one-to-many evaluation approach can be time-consuming as it needs to compute a LLaMA-3-Eval score for *every* ground truth-prediction pair to determine the most appropriate prediction to evaluate against a given ground truth and using many-to-many matching may result in a faster evaluation process. We also show justifications generated by LLaMA-3 for its scores for high and low scoring examples in Table 7 of Appendix C.

Table 6 shows the quantitative results of our experiments. We note that using many-to-many evaluation generally leads to a lower score; however, upon inspection, we find that many-to-many prompt often mismatches a ground truth insight with a wrong prediction (even though a more appropriate prediction was present in the pool of predicted insights). Table 6 shows some mismatches generated by the many-to-many evaluation approach compared to the proposed one-to-many evaluation approach that intuitively does not miss a good prediction if it is present in the list of generated insights.

## 5 RELATED WORK

Our work intersects with the literature on Data Science Benchmarks, LLM Evaluation Frameworks, and Text-to-Analytics Agents.

**Data Science Benchmarks** With enhanced abilities in code generation, utilization of LLM-based data analytics assistants is becoming increasingly prevalent. This has led to the development of numerous data science benchmarks (Lai et al., 2022; Chandel et al., 2022; Zan et al., 2022; Hu et al., 2024; Zhang et al., 2024b). DS-1000 (Lai et al., 2022) focuses on code generation for diverse data science questions sourced from StackOverflow. Similarly, InfiAgent-DABench (Hu et al., 2024) assesses the end-to-end problem-solving ability of LLMs by asking questions based on provided CSV files. DSP (Chandel et al., 2022), evaluates a model's proficiency through a code-infilling task within Jupyter Notebooks. DSEval (Zhang et al., 2024b) focuses on the overall behavior of data science agents without getting into the nuances of code generation techniques. On the other hand, Text2SQL and tabular data reasoning benchmarks assess the ability of models to parse queries, extract information, and synthesize the retrieved data to formulate responses (Katsogiannis-Meimarakis & Koutrika, 2021; Zhong et al., 2018; Chen et al., 2021b). While existing benchmarks primarily evaluate agents on single-step code generation, *InsightBench* shifts the focus to multi-step analysis (Delen & Ram, 2018).

**LLM Evaluation Frameworks** Existing LLM evaluations frameworks focus on handling structured outputs and predominantly rely on pre-formatted prompts to assess code completion (Wu et al., 2023; Zhang et al., 2023a; 2024a; Yao et al., 2023). While recent advancements have seen autonomous agents specializing in intricate data science tasks, including analysis, visualization, and modeling (Qian et al., 2023; 2024; Zhang et al., 2023a), evaluations for these methods often depend on extensive human effort (Cheng et al., 2023) or use more powerful LLMs to assess the output (Dubois et al., 2023; Belyi et al., 2024).

Another work uses the "Capture the Flag" principle, where insights are planted into a dataset as flags to evaluate whether models can uncover them (Laradji et al., 2023). G-Eval (Liu et al., 2023) is a recent technique used to evaluate the quality of free-form texts in terms of factuality and coherence. In this work, we use a variation of G-Eval to score how well the predicted insights are aligned with the ground-truth insight.

**Text-to-Analytics Agents** Chen et al. (2023) explore the application of GPT variants within a data visualization context, highlighting the strengths and limitations of these models. More recent LLM-based data analysis agents include Code Interpreter (OpenAI, 2024) and Pandas Agent (LangChain, 2024) that are capable of processing multiple data formats and answering questions about them. Ma et al. (2023) propose InsightPilot, an advanced automated tool that leverages LLMs to enhance data exploration by automatically identifying goals and generating targeted intentional queries. Vacareanu et al. (2024) showed that LLMs can also perform regression tasks, enhancing their predictive abilities. Additional studies assess the data analysis capabilities of GPT-4 and propose an end-to-end framework for automating data processes (Cheng et al., 2023; Wang et al., 2023; 2024; Hong et al., 2024). Inspired by InsightPilot, we propose *AgentPoirot* that can perform end-to-end data analysis that includes extracting descriptive, diagnostic, predictive, and prescriptive insights.

## 6 CONCLUSION

We have introduced *InsightBench*, a benchmark with 100 diverse datasets that evaluates agents on their ability to perform end-to-end data analytics, including suggesting questions, interpreting answers, and summarizing insights. We ensured each dataset has clear goals and meaningful analysis as ground-truth to evaluate agents reliably. Using LLaMA-3-Eval, an open-source evaluator, we assessed how well the agents extracted insights compare to the ground-truth. We showed how our proposed agent, *AgentPoirot*, can perform data analytics from high-level goals using both open-source and closed-source models and showed that it outperforms existing approaches like Pandas Agent. We believe *InsightBench* will significantly drive advancements in data analytics. For future work, we look into expanding this benchmark to include more categories of data such as healthcare data, social media trends, environmental data, e-commerce analytics, and educational statistics.

## 7 LIMITATIONS

Benchmarks have the risk of reinforcing existing biases or oversimplifying complex decision processes. The design of *InsightBench* necessitates continuous reviews to ensure that the insights generated by agents do not perpetuate biases or lead to misinformed decisions. Additionally, agents' performance on this benchmark should be continually assessed against evolving business practices and technological advancements to maintain relevance and effectiveness. More discussions on limitations are presented in Appendix A.

## 8 REPRODUCIBILITY STATEMENT

To facilitate the reproducibility of our results, we have taken the following measures:

1. Code Availability: All our code, including data used for model implementation, training, and evaluation, is available in the supplementary material.

2. Computing Infrastructure: All LLaMA-3 experiments were conducted on 2 x 80G A100 GPUs.

4. Ablations: A complete list of hyperparameters used in our experiments is provided in Appendix C of the paper. Additionally, all prompts used in this work are included in Appendix D.

5. Random Seeds: To ensure reproducibility, we have run our experiments for 5 random seeds.

6. Evaluation Metrics: We provide a detailed description of all evaluation metrics used in Section 2.2 of the paper. Implementation of these metrics is included in our codebase.

7. Experimental Procedures: Section 3 of the paper includes a comprehensive description of our experimental procedures.

8. Dependencies: A complete list of software dependencies, is provided in the 'requirements.txt' file of the supplementary material.

By specifying these details, we aim to ensure that our work can be readily reproduced and built upon by the research community.

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

## A    LIMITATIONS AND POTENTIAL SOCIETAL IMPACT

*InsightBench* is designed to advance the field of Exploratory Data Analysis (EDA) by providing a challenging framework for evaluating open-ended multi-step data analytics tools and models. The dataset evaluates and helps build conversational data agents with significant potential to enhance decision-making and operational efficiencies in organizations, particularly for end-users who may not have deep technical expertise in data science. Yet, there are important considerations and limitations concerning the current data construction and the benchmark's application in the real-world setting.

**Diversity and Scope of Data Simulation.**    While *InsightBench* covers a spectrum of business areas and themes, the datasets are synthesized based on patterns observed in widely-used ServiceNow. Consequently, the synthetic "flags" are designed to mimic real-world anomalies and trends but may not capture genuine enterprise data's full complexity or unexpected behavior.

**Expanding the Benchmark.**    The current structure of *InsightBench* allows for scalable expansions to include more nuanced scenarios, themes, and more datasets that could involve unstructured data like reports, logs, or emails. Crucial information and actionable insights can also be derived from visual representations such as charts and plots. Further development could also integrate diverse business norms, the ability to analyze the plots and graphs from a visual context alongside language by the agents. Moreover, incorporating feedback from real-world deployments could help refine the datasets to simulate underlying business data better. Lastly, *InsightBench* allows interpreting data types and evaluating outputs for practical utility. Its adaptive framework, which evolves through multi-step interactive feedback, may align with agile development practices. This positions our *InsightBench* a future setup for integrating intelligence into the end-to-end software development process.

## B    APPENDIX: OVERVIEW

Our supplementary material includes the following sections:

- **Section A:** Contains details of data accessibility, generation process, and a breakdown of domains covered by InsightBench.
- **Section B:** Contains results of some ablation experiments and qualitative examples.
- **Section C:** Discusses relevant literature for our work.
- **Section D:** Contains all the prompts used in our experiments.

**Dataset Maintenance**    Authors are committed to ensuring the dataset's regular upkeep and relevance, we will place a system for users to report issues or suggest updates. A feedback form will be available for users to contribute their input. We commit to actively reviewing these suggestions and making the necessary adjustments to the dataset. We also invite contributions from the community through pull requests on InsightBench's GitHub repository.

### B.1    EXAMPLE PROTOCOL FOR DATA GENERATION AND INSIGHT PLANTING

Here, we outline a practical case of how we created a flag dataset for an incidents table with a trend in which the duration of incident resolution increases over time:

1. **Schema Exporting and Standardization:** Initially, the schema for the incidents table is exported and standardized by inspecting a demo instance. Fields such as "number", "opened_at", and "closed_at", among others, are defined.

2. **Data Generation Parameters:** Parameters for generating the data include factors that influence and define the trend. For this example, the parameters include a slope to model the trend in resolution time. Dates ranging from start date to end date are used to set the temporal context of the data. In the experiments section, we ablate and study the effect of the key parameters.

3. **Synthetic Data Creation:** The entire synthetic data generation pipeline is implemented in Python, leveraging libraries such as Pandas for data manipulation and custom scripts for quality control. For example, for fields like IDs, categories of incidents or assets, transaction dates, and status codes, we used random sampling from carefully curated lists of plausible values. The lists were created by analyzing common patterns in actual operational data to capture typical distributions and variances. For example:

- *ID fields:* We generated unique identifiers based on specific formats to simulate real data structures, such as GUIDs or composite keys for hierarchical datasets.

- *Date fields:* Transaction dates and timestamps were randomly sampled within specific ranges, often constrained by meaningful boundaries (e.g., fiscal quarters or business hours) to reflect real-life constraints.

- *Categorical fields:* Categories like incident types and asset classifications were randomly sampled to create a balanced distribution that mirrors real datasets' diversity. Sampling strategies included uniform and weighted distributions to reflect frequency differences in actual usage patterns.

Additionally, we applied statistical validation to ensure that the generated datasets reflected the anticipated distributions and complexities.

4. **Trend Implementation:** The time to resolution is calculated using a linear model where the resolution time increases over the duration of the dataset. The parameter chosen in step 2 is used to model the linear function. The closing time of an incident is determined based on the resolution time and its opening time.

5. **LLM Utilization**: For fields that require diverse and realistic inputs like short descriptions of the incidents, an LLM generates descriptions based on the incident category, ensuring that the data retains an authentic and varied narrative quality.

This approach provides a flexible and scalable framework to produce synthetic datasets that closely resemble real-world business data, allowing us to introduce realistic patterns, variations, and inconsistencies essential for robust model training and evaluation. Further details on our data generation approach, including specific parameters and code snippets, are provided in Appendix B.1.

## B.2 DATA TABLES ACROSS THEMES

Here is an exhaustive and descriptive outline of the topics covered in the benchmark:

- **Incidents Management:** Focuses on tracking, analyzing, and resolving workplace incidents. An extensive description of this data table is discussed afore-mentioned Sec 2.1.1.
- **User Management:** Datasets in this theme are derived from sys_user system table of ServiceNow. This table tracks all user profiles within the platform. It contains information about each user, such as their roles, department affiliations, contact details, and activity status. Key fields include the user's ID, name, email, department, and last login time. This table is used in *InsightBench* for insights related to roles and permissions of employee agents, focusing on schedules and activity patterns.
- **Finance Expenses:** Datasets in this topic examine detailed records of expenses to uncover patterns and optimize budget allocations and prioritization. Datasets are built from fm_expense_list, a system table that logs detailed entries of financial transactions and expenses as part of the financial management module. It includes data points like the expense amount, date, associated user, department, category, and processing status.
- **Inventory Management:** Manages data regarding hardware assets and procurement, aiming to understand patterns in inventory systems. Datasets are derived from alm_hardware, a system table that manages records of all physical assets, particularly IT hardware. Fields in this table detail each asset's tag number assigned to the user, status, location, purchase date, and warranty expiration. It supports asset tracking, lifecycle management, and maintenance activities within an organization.
- **Enterprise Goal Management:** Datasets in this theme evaluate the alignment of departmental performance with overarching goals, focusing on the effectiveness and achievement rates. Datasets are derived from sn_gf_goal system table of ServiceNow, which is essential for evaluating goal management efficiency and aligning departmental outputs with organizational objectives. Fields in this table consist of goal description, start and end dates, current status, owner, priority, and percentage completion.

## B.3 LIST OF DATASETS AND PROBLEMS

Table 2 shows the names of all the datasets in *InsightBench* along with their difficulty and category. Table 3 presents an example of a question-insight pair along with corresponding plots for each category.

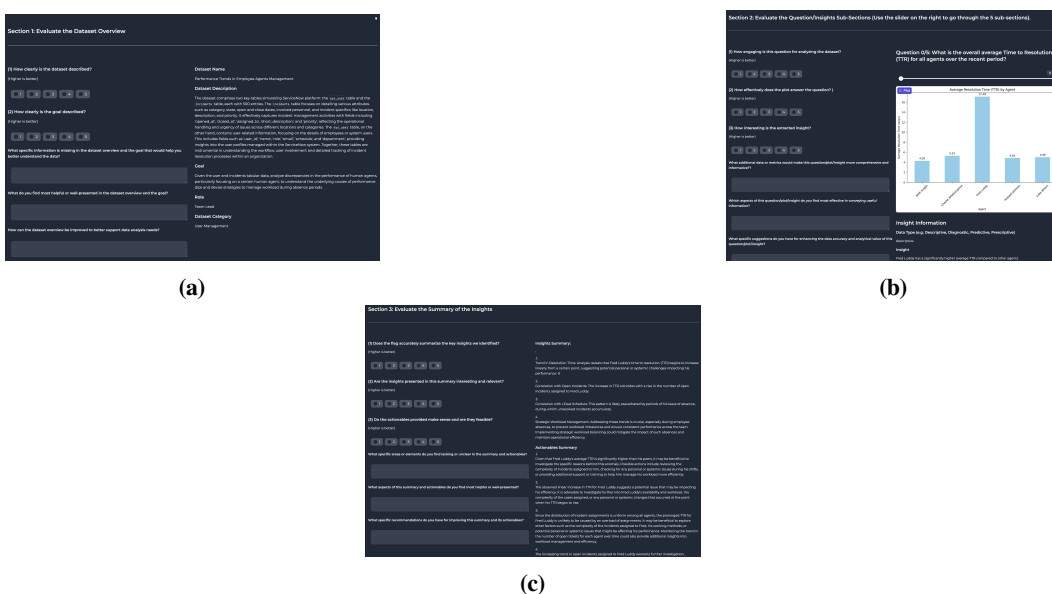

(a)

(b)

(c)

**Figure 5: Screenshots of the Interface used for Dataset Quality-check**:

## B.4 QUALITY-CHECK INTERFACE

To ensure the high standards of data quality and relevance, *InsightBench* has implemented a comprehensive quality-check process involving volunteer contributors. This process is facilitated through a specifically developed gradio interface, which guides the expert reviewers through a structured review of the ground-truth analysis notebooks included in the benchmark (Figure 5). The evaluation is divided into three main sections:

**Section 1: Dataset Overview Evaluation.** Volunteers assess how clearly the dataset and its objectives are described. This includes reviewing the clarity of the dataset description and the specificity and measurability of the stated goals (Figure 5a).

**Section 2: Question and Insights Evaluation.** This section focuses on the engagement and relevance of the questions posed in the notebooks. Volunteers use a slider interface to review each question and corresponding insight sub-sections, evaluating how effectively the plotting code answers the questions and how well the insights align with the analysis goals (Figure 5b).

**Section 3: Summary of Insights Evaluation.** The final section requires volunteers to determine whether the summary accurately encapsulates the key insights and conclusions drawn from the data analysis. This involves checking if the final flag provides a comprehensive and concise summary of the notebook's findings (Figure 5c).

The average rating scores for questions across the three sections is depicted in Figure 6.

## B.5 DETAILS OF THE PROPOSED *AgentPoirot*

Here we describe the proposed baseline *AgentPoirot* in further detail. Figure 7 depicts our pipeline. Initially, *AgentPoirot* uses the dataset and its schema to obtain a set of root questions about the data, aligning with the user's Goal or Role. Each of the root questions can be injected into a Code Generation prompt to obtain and execute code, as well as obtain textual descriptions of the insight found. From these outputs, we generate follow-up questions and start the process again, obtaining a model that navigates the data in depth and in breadth to find interesting insights.

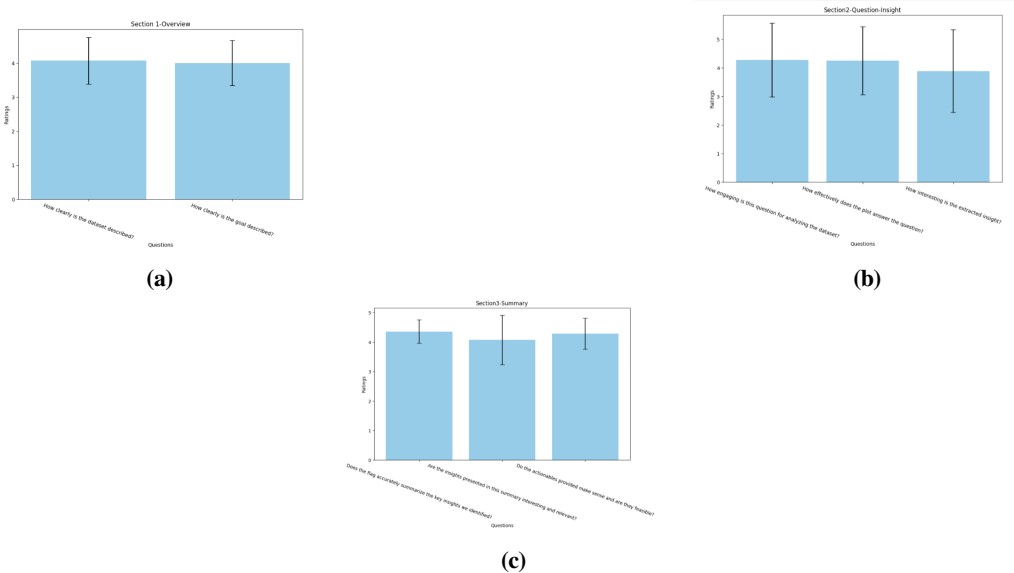

**Figure 6: Distribution of Dataset Quality-check Scores across the three sections**:

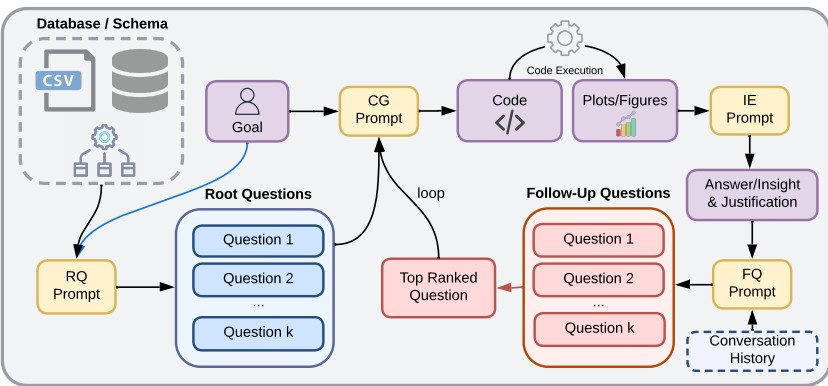

**Figure 7:** *AgentPoirot* uses the "Question Generation (or **QG**) Prompt" (Prompt 1 in Appendix D) that takes as input the dataset schema and the goal to generate $k$ high-level questions. Then, it uses the "Code Generation (or **CG**) Prompt" (Prompt 2 in Appendix D) to generate plots answering the high-level questions. Using the "Insight Extraction (or **IE**) Prompt" (Prompt 5 in Appendix D), and the outputs from the previous step, it generates an *insight* and the *justification* of that insight. Finally, *AgentPoirot* uses the "Follow-Up Question (**FQ**) Prompt" (Prompt 3 in Appendix D) to generate diverse follow-up questions, and selects the top one using the "Question Selection (or **QS**) Prompt" (Prompt 6 in Appendix D). This question is injected back into the **CG** prompt to start the cycle again.

Table 2: **Overview of *InsightBench* Datasets and Problems.** This table enumerates a subset of the first 30 datasets included in the benchmark, detailing their respective topics, difficulty levels, and titles. This enlists diverse scenarios that *InsightBench* covers to evaluate the abilities of data analytics agents.

| Dataset | Dataset ID | Difficulty | Category |
|---|---|---|---|
| Hardware Incident Dataset | 1 | 4 | Incident Management |
| Incident Resolution Time Dataset | 2 | 3 | Incident Management |
| Incident Assignment Distribution Dataset | 3 | 2 | Incident Management |
| Incident Category Trends Over Time | 4 | 4 | Incident Management |
| Time to Resolution Trends Across Incident Categories | 5 | 2 | Incident Management |
| Agent Performance Analysis Over Time | 6 | 4 | Incident Management |
| Incident Assignment and Resolution Efficiency Analysis | 7 | 3 | Incident Management |
| Caller Incident Impact Analysis | 8 | 2 | Incident Management |
| Hardware Incident Analysis During Specific Time Windows | 9 | 4 | Incident Management |
| Incident Resolution Time Trends Analysis | 10 | 3 | Incident Management |
| Category based Incident Trends Analysis | 11 | 4 | Incident Management |
| Hardware Incident Easy Dataset | 12 | 1 | Incident Management |
| User Agent Wellbeing and Incident Volume Analysis | 13 | 2 | Incident Management |
| Performance Trends in Employee Agents Management | 14 | 4 | User Management |
| Workload Distribution and Efficiency Analysis | 15 | 4 | User Management |
| Asset Warranty Analysis | 16 | 2 | Asset Management |
| Asset Cost Analysis by Department | 17 | 3 | Asset Management |
| Asset Warranty and Purchase Date Analysis | 18 | 3 | Asset Management & User Management |
| Expense Management Discrepancies | 19 | 3 | Finance Management |
| Travel Expense Rejection Analysis | 20 | 2 | Finance Management |
| Expense Rejection Trends for New Employees | 21 | 2 | Finance Management & User Management |
| Expense Processing Efficiency Analysis | 22 | 3 | Finance Management |
| Expense Claim Patterns and Fraud Analysis | 23 | 3 | Finance Management |
| Expense Processing Time Analysis | 24 | 3 | Finance Management |
| Expense Processing Dynamics Analysis | 25 | 2 | Finance Management |
| Asset Warranty Analysis | 26 | 2 | Asset Management |
| Management Staffing Analysis in IT Department | 27 | 3 | User Management |
| Goal Achievement Rate Analysis in IT Department | 28 | 2 | Goal Management |
| Goal Management Analysis Category Focus | 29 | 2 | Goal Management |
| Goal Management Analysis in Cost Reduction | 30 | 3 | Goal Management |

Table 3: Instances of insights from *InsightBench* for each category.

| Dataset Category | Insights | Plot |
|---|---|---|
| Incident Management | **Question:** What is the trend in time to resolution (TTR) of incidents across categories? 
 **Insight:** TTR starts to increase linearly for hardware incidents abruptly during a particular time period. |  |
| User Management | **Question:** What is the distribution of reporters per Manager by Department? 
 **Insight:** The average number of reportees per manager in the IT department is significantly higher than other departments. It is 50.5 compared to 8.8 in Customer Support, 11.6 in Finance, 12.8 in HR , and 13.0 in Sales. |  |
| Assets Management | **Question:** What is the relationship between the purchase date of assets and their warranty periods? 
 **Insight:** There is a significant correlation between purchase dates and warranty periods in that the later one purchases an asset the higher the warranty which is a surprising insight. |  |
| Expense Management | **Question:** What is the distribution of processing times of expense reports across different cost brackets? 
 **Insight:** Counter-intuitively, expenses within lower cost brackets experience significantly longer processing times. |  |
| Goal Management | **Question:** How long do goals under 'Cost Reduction' take to achieve? 
 **Insight:** As time passes, there is an increase in the time it takes to achieve 'Cost Reduction' goals. |  |

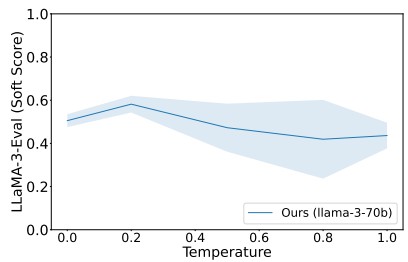 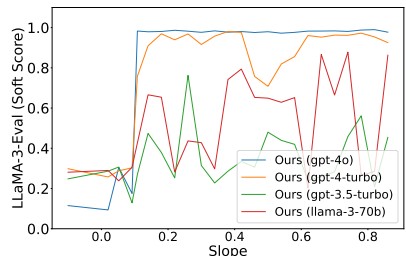

**(a)** Effect of varying sampling temperature (performance reported across the 100 datasets in *InsightBench*)

**(b)** Effect of varying trend intensity (for a specific insight in dataset 2).

**Figure 8:** Sensitivity Analysis.

## C ABLATION RESULTS

Table 4 shows the difference between various evaluation strategies, and Table 6 shows qualitative results for many-to-many and one-to-many evaluation.

**Table 4:** Comparison of different evaluation methods on the first 10 datasets of *InsightBench*.

| Agent | Soft Scores | | | Summary Scores | |
|---|---|---|---|---|---|
| | G-Eval | LLaMA-3-Eval | LLaMA-3-Eval (many-to-many) | G-Eval | LLaMA-3-Eval |
| Ours (gpt-4o) | 0.53 ±0.01 | 0.57 ±0.02 | 0.43 ±0.03 | 0.55 ±0.01 | 0.58 ±0.03 |
| Ours (gpt-4-turbo) | 0.52 ±0.02 | 0.55 ±0.02 | 0.40 ±0.02 | 0.54 ±0.03 | 0.58 ±0.02 |
| Ours (gpt-3.5-turbo) | 0.46 ±0.01 | 0.50 ±0.01 | 0.35 ±0.02 | 0.48 ±0.01 | 0.50 ±0.01 |
| Ours (llama-3-70b) | 0.45 ±0.01 | 0.50 ±0.02 | 0.34 ±0.03 | 0.47 ±0.01 | 0.49 ±0.01 |

**Table 5:** Comparsion of insights generated by PandasAgent and AgentPoirot. Parts **in bold** show why an insight is a close approximation of the ground truth insight. We note that AgentPoirot consistently generates to-the-point insights compared to PandasAgent, which often generates related but slightly off-topic insights.

| Ground Truth Insight | Insight Predicted by PandasAgent (gpt-4o) | Insight Predicted by AgentPoirot (gpt-4o) |
|---|---|---|
| | **Dataset Category: Incident Management** | |
| The time to resolution of incidents is increasing over time | Network assignment group consistently has longer resolution times compared to other groups, with an average resolution time of approximately 1074 hours. | Yes, **there is a clear trend of increasing incident resolution times** over the months, with the average resolution time rising from 113.03 hours in January 2023 to 3150.86 hours in June 2024. |
| The increase in volume of incidents is seen only for one particular category i.e. Hardware | The trend in assigned incidents varies significantly across different assignment groups, with the "Hardware" group having the highest number of incidents (314), followed by "Software" (56), "Network" (48), "Database" (43), and "Service Desk" (39). | The category **'Hardware' has shown the highest increase in the number of incidents** over the given period, with a total increase of 32.0 incidents. |
| | **Dataset Category: User Management** | |
| Fred Luddy has a significantly higher average TTR compared to other agents | There are specific types of incidents where the specified human agent, Fred Luddy, consistently performs worse than others. | The absence of Howard Johnson significantly impacts the workload and performance of other agents, as evidenced by the **high mean resolution times, particularly for Fred Luddy at 410.055 hours**, compared to other agents like Beth Anglin at 102.210 hours. |
| There is no significant change in the distribution of incident categories for Beth Anglin and Luke Wilson during the other agents' PTO | During peak business hours, Luke Wilson and Beth Anglin receive the highest number of incident assignments, while off-peak hours show a more balanced distribution among agents. | No specific periods identified where the number of incidents assigned to agents significantly increases or decreases, and **no correlation with agents' leave periods** has been established. |
| | **Dataset Category: Asset Management** | |
| Servers and Web Servers are the most expensive asset categories on average, followed by computers. | Yes, there are cost outliers within specific model categories as the maximum cost of an asset is 4786, which is significantly higher than the 75th percentile cost of 2834. | Yes, certain model categories such as **'Server' and 'Computer' have a higher variance in asset costs** compared to others, with variances of 4,441,227.02 and 1,063,157.44 respectively, and these categories also have relatively longer mean warranty periods of 730.87 and 1,228.08 days. |
| There is an excessive number of computers per user in HR Department | Yes, in the HR department, the 'Computer' model category has significantly higher costs compared to other models, totaling 61215. | Yes, the **HR department has a disproportionately high number of Computers** (19) compared to other model categories such as Web Servers (5) and Servers (4). |
| | **Dataset Category: Finance Management** | |
| Most of the declined expenses are belong to a few specific users in IT Department | There is a correlation between the user submitting the expense and the likelihood of rejection, as some expenses are declined, indicating a rejection, while others are processed or pending. | Yes, **specific users** such as Vernon Engelman and Helene Iberg f**rom the IT department have higher rates of expense rejections**, particularly in the 'Assets' category. |
| There is a linear positive correlation between new employee's start dates and their expense rejection rates | **There are differences in the approval rates of expense submissions between newer and more experienced employees**, with newer employees having higher rates of submission errors and rejections compared to more experienced employees. | The **rejection rate of expense submissions tends to be higher for employees with shorter employment duration**s, as indicated by the higher rejection rates observed in the initial days of employment. |
| | **Dataset Category: Goal Management** | |
| There is a uniform distribution of goal priorities in the Finance department | There is no significant correlation between the owner of a goal in the Finance department and the time taken for completion. | The **priority level of goals in the Finance department is relatively balanced**, with counts of 58 for Low, 52 for Medium, 52 for High, and 51 for Critical priorities. |
| The IT department exhibits a higher number of both Critical and High priority goals compared to other departments | The Marketing department has the highest number of goals, followed by HR, IT, and Finance. | The **distribution of goal priorities within the IT department** is 56 High, 54 Critical, 10 Low, and 9 Medium, which **shows a higher concentration of High and Critical priorities compared to other departments.** |

**Table 6:** Mismatched examples by many-to-many compared to one-to-many evaluation scheme. Parts **in bold** indicate why the prediction is appropriate to compare against the ground truth insight.

| Ground Truth Insight | Prediction Matched by Many-to-Many Evaluation Scheme | Prediction Matched by One-to-Many Evaluation Scheme |
|---|---|---|
| The time to resolution of incidents is increasing over time | The average resolution time for incidents is 1677.27 hours for '1 - Critical' priority, 1554.63 hours for '2 - High' priority, and 1584.00 hours for '3 - Moderate' priority. | Yes, **there is a clear trend of increasing incident resolution times** over the months, with the average resolution time rising from 113.03 hours in January 2023 to 3150.86 hours in June 2024. |
| The time to resolution of incidents is uniform over time | Beth Anglin has shown the highest efficiency in resolving incidents with a mean resolution time of 153.90 minutes, and her workload has included handling 105 incidents over the given period. | **There are no noticeable patterns** in the types of incidents (categories) that are being resolved more quickly or slowly over time. |
| The volume of Hardware incidents is elevated during specific time windows | Yes, certain 'caller_id' individuals such as 'Don Goodliffe' report more incidents in the 'Hardware' category, and there is a trend showing that 'Bud Richman' and 'David Loo' also frequently report incidents in 'Hardware' and 'Software' categories. | Yes, **there are patterns in the types of incidents reported over time, with 'Hardware' incidents peaking significantly in July and August**. |
| There is a specific agent, Fred Luddy, who is assigned significantly more incidents than others | The assignment group with the highest number of incidents assigned to them is 'Network' with 310 incidents. | **Fred Luddy consistently handles a higher number of incidents compared to others**, with 385 incidents assigned to him. |
| There are higher expense rejection rates for employees with shorter tenure | Yes, the 'Assets' and 'Travel' categories have a higher rate of submission errors or rejections, with counts of 91 and 72 respectively, compared to 'Services' and 'Miscellaneous' which have 16 and 6. | **The rejection rate** of employees **tends to be higher for employees with shorter employment durations**, with many instances of 1.0 rejection rates observed in the initial days of employment, and it gradually decreases as employment duration increases. |
| There are higher success rates in Critical and High priority goals within the IT department | Yes, there are correlations between the metric types and the success rates of projects, as indicated by the plot statistics showing a range of data points for percent complete and target percentage of projects. | The dataset reveals a trend where **projects with 'High' and 'Critical' priority levels generally have higher completion rates**, often exceeding 80%, compared to those with 'Low' and 'Medium' priorities, which show more variability and lower completion rates overall. |

**Table 7:** Comparison of Generated Insights with Ground Truth

| Example | Ground Truth (GT) | Generated Insight | LLaMA-3-Eval Score and Justification |
|---|---|---|---|
| 1 (Good) | There are significantly higher asset costs in the HR Department. | The average cost of assets in the HR department is $4874.25, which is significantly higher compared to the average cost of $1967.26 in other departments. | **Score: 0.81**
The generated text accurately conveys that the HR department has higher asset costs compared to other departments, with specific numerical values supporting the claim. The score is slightly less than 1 because "significant" is used in a general rather than statistical sense. Additional context could further improve this. |
| 2 (Bad) | There is a weak correlation between the number of users and the high cost of computer assets in the HR department. | The distribution of asset types within the HR department includes 19 computers, 5 web servers, and 4 servers, with computers being the most prevalent asset type. | **Score: 0.18**
The generated text does not address the correlation between the number of users and the cost of computer assets. Instead, it discusses asset distribution, which is irrelevant to the GT claim, resulting in a low score. |
| 3 (Bad) | There is a strong positive correlation between the increase in training hours and improved employee performance. | The training program was held over a span of 6 months for six agents. | **Score: 0.11**
The generated text fails to address the correlation between training hours and employee performance. Instead, it provides irrelevant details about the training program's duration and the number of participants, resulting in a low score. |

# D  PROMPT DESIGN

In this section, we illustrate the use of tailored prompts to enhance the analytical capabilities of our models for extracting insights and conducting advanced analytics.

Prompt 1: Prompt used for Extracting Questions from Data

```
SYSTEM MESSAGE:
You the manager of a data science team whose goal is to help stakeholders
    within your company extract actionable insights from their data.
You have access to a team of highly skilled data scientists that can
    answer complex questions about the data.
You call the shots and they do the work.
Your ultimate deliverable is a report that summarizes the findings and
    makes hypothesis for any trend or anomaly that was found.

PROMPT:
### Instruction:

Given the following context:
<context>{context}</context>

Given the following goal:
<goal>{goal}</goal>

Given the following schema:
<schema>{schema}</schema>

Instructions:
* Write a list of questions to be solved by the data scientists in your
    team to explore my data and reach my goal.
* Explore diverse aspects of the data, and ask questions that are
    relevant to my goal.
* You must ask the right questions to surface anything interesting (
    trends, anomalies, etc.)
* Make sure these can realistically be answered based on the data schema.
* The insights that your team will extract will be used to generate a
    report.
* Each question should only have one part, that is a single '?' at the
    end which only require a single answer.
* Do not number the questions.
* You can produce at most {max_questions} questions. Stop generation
    after that.
* Most importantly, each question must be enclosed within <question></
    question> tags. Refer to the example response below:

Example response:
<question>What is the average age of the customers?</question>
<question>What is the distribution of the customers based on their age?</
    question>

### Response:
```

Prompt 2: Prompt used for Generating and Executing code for answering a Question

```
PROMPT:
Given the goal:\n
{goal}

Given the schema:\n
{schema}

Given the data path:\n
{database_path}

Given the list of predefined functions in cba.tools module and their
    example usage:\n\n
{function_docs}

Give me the python code required to answer this question "{question}" and
     put a comment on top of each variable.\n\n

Make a single code block for starting with ```python
Do not produce code blocks for languages other than Python.
Import cba.tools at the beginning.
You must only use the predefined functions mentioned above to make the
    plot.
You must generate one single simple plot and save it as a jpg file.
For the plot, save a stats json file that stores the data of the plot.
For the plot, save a x_axis.json and y_axis.json file that stores a
    maximum of 50 of the most important x and y axis data points of the
    plot, respectively.
Save each json file using the cba.save_json function
For the json file must have a "name", "description", and "value" field
    that describes the data.
The content of the json file should be less than 4500 characters

Call the fix_fnames function in cba.tools at the end of your code.
End your code with ```.

Output code:\n
```

Prompt 3: Prompt used for Generating Diverse Follow-up Questions based on the previous answer

```
SYSTEM:
You the manager of a data science team whose goal is to help stakeholders
     within your company extract actionable insights from their data.
You have access to a team of highly skilled data scientists that can
    answer complex questions about the data.
You call the shots and they do the work.
Your ultimate deliverable is a report that summarizes the findings and
    makes hypothesis for any trend or anomaly that was found.

PROMPT:
Hi, I require the services of your team to help me reach my goal.

<context>{context}</context>

<goal>{goal}</goal>

<schema>{schema}</schema>

<question>{question}</question>

<answer>{answer}</answer>

Instructions:
* Produce a list of follow up questions explore my data and reach my goal
    .
* Note that we have already answered <question> and have the answer at <
    answer>, do not include a question similar to the one above.
* Explore diverse aspects of the data, and ask questions that are
    relevant to my goal.
* You must ask the right questions to surface anything interesting (
    trends, anomalies, etc.)
* Make sure these can realistically be answered based on the data schema.
* The insights that your team will extract will be used to generate a
    report.
* Each question that you produce must be enclosed in <question>content</
    question> tags.
* Each question should only have one part, that is a single '?' at the
    end which only require a single answer.
* Do not number the questions.
* You can produce at most {max_questions} questions.
```

Prompt 4: Prompt used for Generating only one type of Follow-up Questions based on the previous answer

```
SYSTEM:
You the manager of a data science team whose goal is to help stakeholders
    within your company extract actionable insights from their data.
You have access to a team of highly skilled data scientists that can
    answer complex questions about the data.
You call the shots and they do the work.
Your ultimate deliverable is a report that summarizes the findings and
    makes hypothesis for any trend or anomaly that was found.

PROMPT:
Hi, I require the services of your team to help me reach my goal.

<context>{context}</context>

<goal>{goal}</goal>

<schema>{schema}</schema>

<question_type>{question_type}</question_type>

<question>{question}</question>

<answer>{answer}</answer>

Instructions:
* Produce a list of follow-up questions to explore my data and reach my
    goal.
* Note that we have already answered <question> and have the answer at <
    answer>, do not include a question similar to the one above.
* Explore diverse aspects of the data, and ask questions that are
    relevant to my goal.
* You must ask the right questions to surface anything interesting (
    trends, anomalies, etc.)
* Make sure these can realistically be answered based on the data schema.
* The insights that your team will extract will be used to generate a
    report.
* The question has to adhere to the type of question that is provided in
    the <question_type> tag
* The type of question is either descriptive, diagnostic, prescriptive,
    or predictive.
* Each question that you produce must be enclosed in <question>content</
    question> tags.
* Each question should only have one part, that is a single '?' at the
    end which only require a single answer.
* Do not number the questions.
* You can produce at most {max_questions} questions.
```

Prompt 5: Prompt used for Interpreting the Solution

```
PROMPT:
### Instruction:
You are trying to answer a question based on information provided by a
    data scientist.

Given the context:
<context>
    You need to answer a question based on information provided by a data
     scientist.
</context>

Given the goal:
<goal>{goal}</goal>

Given the question:
<question>{question}</question>

Given the analysis:
<analysis>
    <message>
        {message}
    </message>
    {insights}
</analysis>

Instructions:
* Based on the analysis and other information provided above, write an
    answer to the question enclosed with <question></question> tags.
* The answer should be a single sentence, but it should not be too high
    level and should include the key details from justification.
* Write your answer in HTML-like tags, enclosing the answer between <
    answer></answer> tags, followed by a justification between <
    justification></justification> tags.
* Refer to the following example response for the format of the answer
    and justification.

Example response:
<answer>This is a sample answer</answer>
<justification>This is a sample justification</justification>

### Response:
```

Prompt 6: Prompt used for Selecting the best Question based on previously asked questions and the goal

```
SYSTEM MESSAGE:
You the manager of a data science team whose goal is to help stakeholders
     within your company extract actionable insights from their data.
You have access to a team of highly skilled data scientists that can
    answer complex questions about the data.
You call the shots and they do the work.
Your ultimate deliverable is a report that summarizes the findings and
    makes hypothesis for any trend or anomaly that was found.

PROMPT:
Hi, I require the services of your team to help me reach my goal.

<context>{context}</context>

<goal>{goal}</goal>

<prev_questions>{prev_questions_formatted}</prev_questions>

<followup_questions>{followup_questions_formatted}</followup_questions>

Instructions:
* Given a context and a goal, select one follow up question from the
    above list to explore after prev_question that will help me reach my
    goal.
* Do not select a question similar to the prev_questions above.
* Output only the index of the question in your response inside <
    question_id></question_id> tag.
* The output questions id must be 0-indexed.
"""
```

Prompt 7: Prompt used for Summarizing the Insights

```
SYSTEM_MESSAGE:
You the manager of a data science team whose goal is to help stakeholders
    within your company extract actionable insights from their data.
You have access to a team of highly skilled data scientists that can
    answer complex questions about the data.
You call the shots and they do the work.
Your ultimate deliverable is a report that summarizes the findings and
    makes hypothesis for any trend or anomaly that was found.

PROMPT:
Hi, I require the services of your team to help me reach my goal.

<context>{context}</context>

<goal>{goal}</goal>

<history>{history}</history>

Instructions:
* Given a context and a goal, and all the history of <question_i><
    answer_i> pairs from the above list generate the 3 top insights that
    will help me reach my goal.
* Output each insight within this tag <insight></insight>.
```

Prompt 8: Prompt used for generating the synthetic data for non-controllable elements of an incidents dataset.

```
gpt_fields = ["short_description", "assignment_group"]

field_info = "\n".join(
    [
        "<field>"
        + f"<name>{f}</name><type>{col_info[f]['type']}</type>"
        + (
            f"<choices>{col_info[f]['choices']}</choices>"
            if "choices" in col_info[f]
            else ""
        )
        + "</field>"
        for f in gpt_fields
    ]
)

prompt = f"""
Create a fake ServiceNow incident by generating realistic values for the
    following fields:
<fields>
    {gpt_fields}
</fields>

Here are the types of each of these fields and the permitted values:
<field_info>
    {field_info}
</field_info>

Produce a json dictionnary with a single value for each of these fields.
Make sure to respect allowed values and use their full diversity.

Example:
{{"short_description": "My laptop is broken"}}

Respond with the json only.

"""

valid_data = []
while len(valid_data) < num_samples:
    try:
        completion = client.chat.completions.create(
            model="gpt-4-turbo",
            messages=[
                {
                    "role": "system",
                    "content": "You're going to help me generate
    simulated data that looks like it really came from ServiceNow Glide
    tables.",
                },
                {"role": "user", "content": prompt},
            ],
        )
        incident = json.loads(completion.choices[0].message.content)

        # Validate values
        assert set(gpt_fields) == set(incident.keys())
        for f in gpt_fields:
            if "choices" in col_info[f]:
                assert incident[f] in col_info[f]["choices"]
```

```
        # Accept it
        valid_data.append(incident)
        print(f"... {len(valid_data)} / {num_samples}")

    except:
        print("... invalid value, trying again")
```

Prompt 9: Prompt used for LLaMA-3-Eval Scores

```
SYSTEM:
You are a high school teacher evaluating student responses to a question.
    You are tasked with grading the response based on how well it
    answers the question. You are to provide a numerical rating for how
    well the response answers the question based on the ground truth
    answer.

PROMPT:
Below is an instruction that describes a task. Write a response that
    appropriately completes the request.

### Instruction:
Provided Answer:
{answer}

Ground Truth Answer:
{gt_answer}

Follow these instructions when writing your response:
* On a scale of 1-10, provide a numerical rating for how close the
    provided answer is to the ground truth answer, with 10 denoting that
    the provided answer is the same as ground truth answer.
* Your response should contain only the numerical rating. DONOT include
    anything else like the provided answer, the ground truth answer, or
    an explanation of your rating scale in your response.
* Wrap your numerical rating inside <rating></rating> tags.
* Check very carefully before answering.
* Follow the output format as shown in the example below:
Example response:
<rating>7</rating>

### Response:
```

Prompt 10: Prompt used for many-to-many LLaMA-3-Eval Scores

```
SYSTEM:
You are a high school teacher evaluating student responses to some
    questions. Before scoring their answers, you need to first match each
     ground truth answer with the most appropriate answer provided by the
     student.

PROMPT:
Below is an instruction that describes a task. Write a response that
    appropriately completes the request.

### Instruction:
Predicted Answers:
{pred_list}

Grouth Truth Answers:
{gt_list}

For each ground truth answer above, provide the index of the most
    appropriate predicted answer (1-indexed).
Each line must contain a single integer value denoting the id of the
    matched prediction.
If there is no appropriate prediction for a ground truth answer, write
    -1.
Check very carefully before answering.

### Response:
```

