# OpenReview forum: "InsightBench: Evaluating Business Analytics Agents Through Multi-Step Insight Generation"
_ICLR.cc/2025/Conference — ICLR 2025 Poster_

### Official Review · Reviewer_FGNL · 2024-11-04

**Soundness:** 3
**Presentation:** 4
**Contribution:** 4
**Rating:** 8
**Confidence:** 4

**Summary:**

This paper presents a new benchmark dataset called InsightBench, which evaluates end-to-end analytics tasks' performance in real-world enterprise troubleshooting. The benchmark includes 100 datasets extracted from the ServiceNow platform. The current benchmarks mainly focus on single round-trip query-answer evaluations. This proposed benchmark aims to evaluate an agent's holistic problem-solving skills.

In addition to proposing a new benchmark dataset, the authors also developed a new baseline agent called AgentPoirot, which demonstrated superior performance compared to existing popular agents such as Pandas Agent.

**Strengths:**

1) This paper is a leap forward in laying a foundation for building more robust enterprise-grade business analytics agents. I agree that the current benchmarks primarily just focus on simple code completion or single-query tasks. Real-world problems almost always involve multi-step reasoning and insight development.

2) The benchmark building process is very well-documented and detailed, which was a huge plus from me. It was also clever to use the real-world data schema from a popular troubleshooting ticketing platform, ServiceNow, which ensures the applicability of this approach.

3) Using LLaMA-3-Eval as an alternative to GPT-4 which is a closed API. For the sake of science, using LLaMa is a huge boost in transparency.

4) Plus, AgentPoirot showed impressive performance against Pandas Agent in end-to-end analysis. This result shows that properly designed benchmarks can drive the development of superior agents.

5) The quality assurance process using both expert reviewers and extensive validation was impressive.

**Weaknesses:**

1) Whether synthetic data represents the real-world context is questionable. The synthetic data generation part is less well-documented than other parts of the paper. The authors should explain better how their synthetic data generation process works and how they ensured the representativeness of the synthetic data.

2) The authors mentioned the four types of analytics up front (descriptive, diagnostic, predictive, and prescriptive). However, this notion of four types of analytics was not as prevalent in the later part, where the results were presented. Consider looping back to these four types of analytics when presenting the results and performance.

3) I understand Pandas Agent is probably the most popular agent aiming to achieve similar goals. However, I would like to see the comparison against some other agents, if there are any.

4) This paper focuses too much on showing the strengths of the benchmark and the specialized agent, Agent Poirot. Providing the boundary conditions and limits where the agent fails would be helpful for future studies in this line.

**Questions:**

1) Do you have any evidence that the distribution of your data (for example, insight types--incident, user, asset, finance, goal management) resembles real-world business scenarios?

2) Agent Poirot works best with a temperature of 0.2. This value looks interesting to me. Do you know why the agent works best at such a low temperature?

3) It was encouraging to see that LLaMa-3-Eval produced consistent outcomes compared to G-Eval, which is known to be well aligned with human judgments. Was there any way to evaluate whether LLaMa-3-Eval output is also directly aligned with human judgments?

**Details Of Ethics Concerns:**

If it involves the real-world actual ServiceNow data, it might need to be anonymized, etc. I think the authors did proper steps to clear off this issue, but I just wanted to double check. Plus, the authors need to double check whether IRB approval is required. If the ServiceNow platform was only used for the structure of the benchmark dataset, I don't think there's any need for ethics review.

---

> ### Author Response · Authors · 2024-11-20
> **Rebuttal (1/3) for Reviewer FGNL on InsightBench**
>
> Thank you for recognizing this work as a key step toward robust enterprise-grade analytics agents, with a focus on real-world, multi-step reasoning and practical benchmarks using ServiceNow data. We also appreciate your acknowledgment of LLaMA-3-Eval’s transparency, AgentPoirot’s strong performance, and our rigorous quality assurance process.
>
> Below we address your concerns.
>
> **1) Whether synthetic data represents the real-world context is questionable. The synthetic data generation part is less well-documented than other parts of the paper. The authors should explain better how their synthetic data generation process works and how they ensured the representativeness of the synthetic data.**
>
> - Our synthetic data was vetted by 12 ServiceNow experts that we partnered with that have deep knowledge of actual enterprise data to ensure it accurately represents real-world nuances.  Additionally, we have conducted a human evaluation study (detailed in Appendix B.4) where enterprise data specialists rated our synthetic data highly for its quality, coverage, and relevance.
>
> - We have found consistent relative performance results (w.r.t. L3-Eval metric) between AgentPoirot and PandasAgent on both our synthetic data and real ServiceNow datasets. The real-world data includes CSM, ITSM, and HR.
>
> |        Method    | Our Synthetic Data | Real World Data |
> |--------------------|---------------------------|------------------------|
> | PandasAgent |           0.54               |          0.58           |
> |  AgentPoirot   |           0.60               |         0.64            |
>
> - We unfortunately can’t include real enterprise data due to the proprietary nature of the data.
> - We have updated Section 2.1.3 and Appendix B.1 of the paper to include more details to clarify the synthetic data generation.
>
> **2) The authors mentioned the four types of analytics up front (descriptive, diagnostic, predictive, and prescriptive). However, this notion of four types of analytics was not as prevalent in the later part, where the results were presented.**
>
> - Thank you for pointing this out. Figure 4 now includes performance breakdowns by insight types. As shown in Figure 4c, agent performance decreases progressively from Descriptive insights to Diagnostic, Prescriptive, and Predictive insights, which reflects the increasing complexity of these analytics types.
>
> **3) Do you have any evidence that the distribution of your data (for example, insight types--incident, user, asset, finance, goal management) resembles real-world business scenarios?**
>
> - The list of possible values for various columns (such as insight types, ID types, and dataset categories) was carefully designed by analyzing statistical patterns observed in real-world enterprise data, particularly ServiceNow data. As a result, the distributions closely reflect real-world, realistic business scenarios.
>
> **4) Agent Poirot works best with a temperature of 0.2. This value looks interesting to me. Do you know why the agent works best at such a low temperature?**
>
> - We believe a temperature setting of 0.2 strikes an effective balance between accurate mathematical inference and controlled freeform insight extraction. Lower temperatures encourage more deterministic outputs, which enhances the precision needed for reliable quantitative analysis and minimizes unnecessary variability in responses. This helps Agent Poirot maintain a focus on accurate code generation while still allowing some flexibility for generating insightful observations.
>
> (1/3) please see next comment for the rest of the rebuttal

---

> > ### Author Response · Authors · 2024-11-20
> > **Rebuttal (2/3) for Reviewer FGNL on InsightBench**
> >
> > **5) I understand Pandas Agent is probably the most popular agent aiming to achieve similar goals. However, I would like to see the comparison against some other agents, if there are any.**
> >
> > - Below, we present the results compared to PowerBI's Copilot as an additional agent for extracting insights, and we obtained the following results:
> >
> > | Method | L3-Eval (Insight-level) | L3-Eval (Summary-level) |
> >  |----------------|-------------------------|-------------------------|
> > | **PowerBI** | 0.28 | 0.24 |
> > | **AgentPoirot**| 0.60 | 0.41 |
> >
> > - PowerBI generated low-quality insights and often has inconsistent output formats for the same prompt. Below are some generated examples. We see that PowerBI often just outputs metric values, instead of a more insightful answer.
> >
> > | Insight Type                 | PowerBI Insight                                                                 | AgentPoirot Insight                                                                                          |
> > |------------------------------|---------------------------------------------------------------------------------|-------------------------------------------------------------------------------------------------------------|
> > | **Average Resolution Time**  | Average resolution time for incidents: 181.88 hours                            | There is a clear trend of increasing incident resolution times over the months, rising from 113.03 hours in January 2023 to 3150.86 hours in June 2024. |
> > | **Incident Distribution**    | Number of incidents per assignment group: {‘Network’: 179, ‘Database’: 170, ‘Software’: 139, ‘Hardware’: 127, ‘Service Desk’: 85} | The category ‘Network’ has the highest number of incidents over the given period (179), followed by ‘Database’ (170), ‘Software’ (139), ‘Hardware’ (127), and ‘Service Desk’ (85). |
> >
> > - We have conducted a thorough literature review to find the strongest baseline available. PandasAgent, which leverages the latest data analytics strategies was found to be the most popular. We also considered the following:
> >   - Microsoft’s Insightpilot was a potential choice, but neither the code nor the prompts are available, so we couldn’t use it.
> >   - Data-Copilot is limited to Chinese financial data, which isn’t suitable for our task.
> >   - OpenAgents was also considered, but the public version only allows 10 requests, which isn’t enough for meaningful benchmarking.
> >   - InfiAgent-Dabench is available, but we excluded it due to its extremely poor performance in this benchmark. It handles only simple tasks like “Generate Python code to compute the mean of a list,” but struggles with more complex queries like “What is the distribution of incidents by category in this dataset?”
> >
> > **6) This paper focuses too much on showing the strengths of the benchmark and the specialized agent, Agent Poirot. Providing the boundary conditions and limits where the agent fails would be helpful for future studies in this line.**
> >
> > - Most observed failure cases occur when the trend in the data is too subtle, leading the agent to output statements like, “There is no clear trend between XX and YY,” even when a weak trend exists.
> >
> > - Additionally, the agent can generate insights based on incorrect entities, as illustrated in the example below:
> >    - **Example (for poorly generated insights):**
> >        - **Ground Truth (GT):** There is a weak correlation between the number of users and the high cost of computer assets in the HR department.
> >        - **Generated:** The distribution of asset types within the HR department includes 19 computers, 5 web servers, and 4 servers, with computers being the most prevalent asset type.
> >        - **LLaMA-3-Eval Score:** 0.18
> >        - In this case, the agent focuses on asset distribution rather than addressing the cost correlation specified in the ground truth. These limitations suggest that while the agent performs well with clear patterns, subtle or nuanced trends may not always be captured accurately, and misinterpretations of target entities can occur.
> >
> > - We also have a set of 100 datasets for sensitivity analysis where we vary the trend intensities to evaluate what trend intensity is needed for the agent to identify it (see Figure 8b).
> >
> > (2/3) please see next comment for the rest of the rebuttal

---

> ### Author Response · Authors · 2024-11-20
> **Rebuttal (3/3) for Reviewer FGNL on InsightBench**
>
> **7) Privacy, security and safety**
>
> - No real ServiceNow data was used, so there’s no risk of exposing personal information. The dataset was only based on the platform’s structure, so anonymization or IRB approval wasn’t needed.
>
> **8) It was encouraging to see that LLaMa-3-Eval produced consistent outcomes compared to G-Eval, which is known to be well aligned with human judgments. Was there any way to evaluate whether LLaMa-3-Eval output is also directly aligned with human judgments**
>
> - While LLaMA-3-Eval scores' consistency with G-Eval indicates that it also aligns with human judgement to at least some extent, we investigated example interpretations of LLaMA-3-Eval below for further evidence of human alignment (also added to the paper in Appendix C, Table 7):
>
>   - **Example 1 (for good generated insights):**
>      - **Ground Truth (GT):** There are significantly higher asset costs in the HR Department.
>      - **Generated:** The average cost of assets in the HR department is \\$4874.25, which is significantly higher compared to the average cost of \\$1967.26 in other departments.
>      - **LLaMA-3-Eval Score:** 0.81
>     - **Justification:** The generated text accurately conveys that the HR department has higher asset costs compared to other departments, with specific numerical values supporting the claim. The score is slightly less than 1 because the term "significant" is used in a general sense rather than in a statistical context. The generated text could be improved by providing additional context for the high asset costs.
>   - **Example 2 (for poorly generated insights):**
>       - **Ground Truth (GT):** There is a weak correlation between the number of users and the high cost of computer assets in the HR department.
>       - **Generated:** The distribution of asset types within the HR department includes 19 computers, 5 web servers, and 4 servers, with computers being the most prevalent asset type.
>       - **LLaMA-3-Eval Score:** 0.18
>       - **Justification:** The generated text does not address the correlation between the number of users and the cost of computer assets, which is the key point of the ground truth. Instead, it discusses the distribution of asset types, which is irrelevant to the claim, resulting in a low score.
>
> - These examples highlight how LLaMA-3-Eval provides detailed and interpretable feedback, with scores reflecting both the relevance and accuracy of the generated insights.
>
> (3/3) End of Rebuttal

---

### Official Review · Reviewer_nFSq · 2024-11-04

**Soundness:** 3
**Presentation:** 3
**Contribution:** 3
**Rating:** 3
**Confidence:** 2

**Summary:**

InsightBench is a new way to test how well AI language models can handle complex business analysis tasks. Unlike older tests that only look at simple questions, this one checks if AI can do a whole analysis from start to finish - asking questions, finding important information, and suggesting what to do next.

The creators made 100 different datasets about things like finance and managing problems. Some of the data is made up, and some is real, to make it feel like actual business situations. They also came up with their own AI agent called AgentPoirot, which did better than other AI tools at figuring out different kinds of insights from the data.

To make sure anyone can use and improve on their work, the team used an open-source AI called LLaMA-3 to judge how well the AI agents do. They hope this will help push forward the development of AI that can do complete data analysis for businesses without needing humans to guide it every step of the way.

**Strengths:**

1. Diverse coverage: The dataset covers diverse business domains like finance and incident management, so can be widely useful, although one can argue that the dataset is not focused on a particular domain.
1. Quality: The dataset insights are evaluated by experts, hence there is an assurance of quality. The two way evaluation (summary + insights) will also help users of the platform be more confident in the results.

**Weaknesses:**

1. Synthetic data: This is a long standing tussle for evaluation -- how realistic the data can get. InsightBench datasets depend on synthetic data generation which can make the data quality lower as the data regresses from real world scenarios. Particularly, it is unclear how well the data represents the unhappy paths / scenarios often found in real world. One suggestion might be to further split up the datasets into Happy path and Challenge datasets, with challenge datasets using less synthetic data, and being more realistic of real world scenarios.
1. Dependency on structure: InsightBench focuses on structured, tabular data, but increasingly other data formats like unstructured text, images, or complex graphs are becoming relevant for business analytics.
1. Scale: It is unclear how the dataset can be kept up to date with evolving business tasks, how quality will be maintained, how new domains will be added etc.

**Questions:**

1. Synthetic data
   * How do you ensure that InsightBench datasets capture the complexity of real-world enterprise data?
   * Could you elaborate on the methods used to simulate unpredictable or "noisy" data patterns?
   * Have you incorporated real-world anomalies or edge cases into your data generation process?
   * Have you validated the synthetic data against any real-world datasets?
1. Dependency on structure
   * Do you have plans to incorporate unstructured data (like text logs or emails) that is common in businesses?
1. Scale
   * Do you have a process in place for regularly updating the dataset, or have you considered partnering with industry experts to ensure the dataset remains relevant to current business practices?

---

> ### Author Response · Authors · 2024-11-20
> **Rebuttal (1/2) for Reviewer nFSq on InsightBench**
>
> We appreciate the reviewer’s recognition of the dataset's coverage across diverse business domains like finance and incident management, making it broadly useful even if not focused on a single domain. We also thank the reviewer for noting the expert-evaluated insights and two-way evaluation, which help build user confidence in the results.
>
> Below we address each concern:
>
> **1) InsightBench datasets depend on synthetic data generation which can make the data quality lower as the data regresses from real world scenarios. Particularly, it is unclear how well the data represents the unhappy paths / scenarios often found in real world. One suggestion might be to further split up the datasets into Happy path and Challenge datasets , with challenge datasets using less synthetic data, and being more realistic of real world scenarios. How do you ensure that InsightBench datasets capture the complexity of real-world enterprise data?**
>
> - Our synthetic data was vetted by 12 ServiceNow experts that we partnered with that have deep knowledge of actual enterprise data to ensure it accurately represents real-world nuances.  Additionally, we have conducted a human evaluation study (detailed in Appendix B.4) where enterprise data specialists rated our synthetic data highly for its quality, coverage, and relevance.
>
> - The domain experts also helped ensure that the injected trends reflect the “unhappy”, “happy”, and “challenge” paths that occur in the real world. In that regard, our benchmark includes datasets at varying difficulty levels (easy, medium, hard) to represent a range of scenarios, including those with subtle or hard-to-detect trends. We conducted user studies with expert data scientists where they managed to identify more trends in "easy" datasets and fewer in harder ones, helping us assess the difficulty levels where current analytics succeed or fail.
>
> - While privacy concerns prevent us from including actual real data, we are committed to collaborating with domain experts to ensure the datasets remain high-quality and realistic as the benchmark evolves.
>
> **2) InsightBench focuses on structured, tabular data, but increasingly other data formats like unstructured text, images, or complex graphs are becoming relevant for business analytics. Do you have plans to incorporate unstructured data (like text logs or emails) that is common in businesses?**
>
> - Unstructured data, like text, images, and graphs, is a very interesting area for business analytics, but it is beyond the scope of InsightBench and it is something we are working on as future versions of this work. We mention this as limitation in Appendix A.
>
> - Our focus with InsightBench is on structured, tabular data because it is still a core part of many data analytics applications and decision-making processes that is far from being solved. This version of InsightBench is already steps ahead of existing benchmarks that focus on closed-form, templated answer evaluations.
>
> **3) Do you have a process in place for regularly updating the dataset, or have you considered partnering with industry experts to ensure the dataset remains relevant to current business practices?**
>
> - Our data generation pipeline is designed to be flexible and scalable, allowing us (with the help of domain experts) to easily generate new data for various business tasks by following the methodology outlined in the paper.
>
> - We also plan to open-source the benchmark and actively collaborate with the broader community. This approach will help keep the benchmark up-to-date, incorporate new types of trends and insights, and maintain high-quality standards over time.
>
> - We have partnered with leading enterprises like ServiceNow to ensure the quality and realism of the data, and we aim to expand our partnership to more enterprises in the future.
>
> **4) Could you elaborate on the methods used to simulate unpredictable or "noisy" data patterns?**
>
> - We use a systematic approach to simulate both “noisy” and “clean” trends, as detailed in Section 2.1.2 and Appendix B.1.
> - We have included 15 datasets in our benchmark without dominant trends to evaluate whether an agent would produce false positives by generating insights where none exist. On these datasets, AgentPoirot scored an L3-Eval of 0.864, compared to PandasAgent’s 0.821, showing that AgentPoirot is better at avoiding false insights.
> - Specifically, the agent generates insights like "There are no insights; nothing specific to analyze”, “There are visible trends in [Quantity XX]”, and “The dataset is missing [Quantity YY] required for analysis.” in those cases.
> - We also included a set of 100 datasets for sensitivity analysis where we vary the trend intensities to evaluate what trend intensity is needed for the agent to identify it.
>
> (1/2) please see next comment for the rest of the rebuttal

---

> > ### Author Response · Authors · 2024-11-20
> > **Rebuttal (2/2) for Reviewer nFSq on InsightBench**
> >
> > **5) Have you incorporated real-world anomalies or edge cases into your data generation process?**
> >
> > - Yes, the injected trends mostly reflect real-world anomalies and edge cases, incorporated with guidance from domain experts, including complex seasonal patterns.
> >
> > - For example, in Datasets 14 and 6, the average Time to Resolution (TTR) steadily increases despite adequate staffing. The LLM Agent must analyze further to uncover that this isn’t due to ineffective support staff but rather because some staff are on leave, causing others to be overloaded.
> >
> > **6) Have you validated the synthetic data against any real-world datasets?**
> >
> > - We have evaluated on real ServiceNow datasets and we have found consistent relative performance results (w.r.t. L3-Eval metric) between AgentPoirot and PandasAgent on both our synthetic and real world data. The real-world data includes CSM, ITSM, and HR.
> >
> > |        Method    | Our Synthetic Data | Real World Data |
> > |--------------------|---------------------------|------------------------|
> > | PandasAgent |           0.54               |          0.58           |
> > |  AgentPoirot   |           0.60               |          0.64           |
> >
> > - We unfortunately cannot include real enterprise data due to its proprietary nature.
> > - Please note that our synthetic data was vetted by 12 ServiceNow experts that we partnered with that have deep knowledge of actual enterprise data to ensure it accurately represents real-world nuances.  Additionally, we have conducted a human evaluation study (detailed in Appendix B.4) where enterprise data specialists rated our synthetic data highly for its quality, coverage, and relevance.
> >
> >
> > (2/2) End of Rebuttal

---

> > > ### Author Response · Authors · 2024-11-25
> > > **Rebuttal Follow-Up for InsightBench**
> > >
> > > Dear Reviewer, thank you again for your detailed feedback.
> > >
> > > Given the limited discussion period, we kindly ask you to review and respond to our rebuttal, where we believe have addressed all your concerns. If anything is unclear, please don’t hesitate to let us know, we’d be happy to clarify further.
> > >
> > > We appreciate your time and insights.

---

### Official Review · Reviewer_DA8a · 2024-11-05

**Soundness:** 3
**Presentation:** 4
**Contribution:** 4
**Rating:** 6
**Confidence:** 3

**Summary:**

This paper introduces InsightBench, a benchmark dataset designed to evaluate the performance of data analytics agents in generating multi-step insights from business data. The authors propose AgentPoirot, a new agent model that outperforms existing tools such as the Pandas Agent. The benchmark dataset includes 100 datasets across various business themes, each with planted insights for the agents to identify. A two-way evaluation mechanism using LLaMA-3 is implemented to assess agent performance.

**Strengths:**

· Originality: The introduction of InsightBench is a significant innovation in the evaluation of data analytics agents. Unlike previous benchmarks, InsightBench focuses on multi-step analytics, providing a comprehensive framework for assessing agents' abilities to perform holistic analysis rather than isolated tasks.

· Clarity: The paper is clearly written, with a logical progression from motivation to methodology and results. Figures and tables effectively support the text, particularly in explaining how InsightBench differs from existing single-step benchmarks.

· Significance: InsightBench is a meaningful contribution that could substantially impact the development of data analytics agents. By offering a platform for evaluating agents' end-to-end performance, InsightBench encourages the creation of more versatile and context-aware analytics models, filling a notable gap in the data analytics research community.

**Weaknesses:**

· Limitations in Real-World Validation: While InsightBench provides synthetic data and insights, the effectiveness of AgentPoirot and other evaluated agents in handling real-world, noisy datasets remains untested. To demonstrate its practical value, further evaluation on real-world data (beyond synthetic datasets) would enhance the relevance of InsightBench.

· Interpretability of LLaMA-3-Eval Scores: The LLaMA-3-Eval scoring method could benefit from additional explanation regarding its interpretability and stability, particularly in cases where scores may fluctuate with subtle prompt changes.

**Questions:**

1. How does InsightBench handle noisy or incomplete data? While the synthetic datasets are well-curated, it would be valuable to know how robust InsightBench and AgentPoirot are to missing data or noise, as these are common in real-world applications.

2. What is the scalability of InsightBench for larger datasets? Since InsightBench datasets are capped at 500 entries, how would the benchmark scale to larger, more complex datasets? Would AgentPoirot's performance degrade with larger data volumes?

3. Are there plans to incorporate real-world data into InsightBench? A dataset that mixes synthetic and real-world data could potentially improve the benchmark’s applicability to practical scenarios.

---

> ### Author Response · Authors · 2024-11-20
> **Rebuttal (1/2) for Reviewer DA8a on InsightBench**
>
> We appreciate the reviewer’s recognition of InsightBench as an innovative, clear, and impactful benchmark for multi-step analytics, promoting versatile and context-aware models while addressing key gaps in data analytics research.
>
> Below we address each concern:
>
> **1) While InsightBench provides synthetic data and insights, the effectiveness of AgentPoirot and other evaluated agents in handling real-world, noisy datasets remains untested.**
>
> - We have evaluated on real ServiceNow datasets and we have found consistent relative performance results (w.r.t. L3-Eval metric) between AgentPoirot and PandasAgent on both our synthetic and real world data. The real-world data includes CSM, ITSM, and HR.
>
> |        Method    | Our Synthetic Data | Real World Data |
> |--------------------|---------------------------|------------------------|
> | PandasAgent |           0.54               |          0.58           |
> |  AgentPoirot   |           0.60               |          0.64           |
>
> - We unfortunately can’t include real enterprise data due to the proprietary nature of the data.
> - Please note that our synthetic data was vetted by 12 ServiceNow experts that we partnered with that have deep knowledge of actual enterprise data to ensure it accurately represents real-world nuances.  Additionally, we have conducted a human evaluation study (detailed in Appendix B.4) where enterprise data specialists rated our synthetic data highly for its quality, coverage, and relevance.
>
> **2) How does InsightBench handle noisy or incomplete data?**
>
> - We have included 15 datasets in our benchmark without dominant trends to evaluate whether an agent would produce false positives by generating insights where none exist. On these datasets, AgentPoirot scored an L3-Eval of 0.864, compared to PandasAgent’s 0.821, showing that AgentPoirot is better at avoiding false insights.
>
> - Specifically, the agent generates insights like "There are no insights; nothing specific to analyze”, “There are visible trends in [Quantity XX]”, and “The dataset is missing [Quantity YY] required for analysis.” in those cases.
>
> - We also have a set of 100 datasets for sensitivity analysis where we vary the trend intensities to evaluate what trend intensity is needed for the agent to identify it (see Figure 8b).
>
> **3) What is the scalability of InsightBench for larger datasets? Since InsightBench datasets are capped at 500 entries, how would the benchmark scale to larger, more complex datasets? Would AgentPoirot's performance degrade with larger data volumes?**
>
> - We have evaluated AgentPoirot on real-world ServiceNow datasets with 8000+ rows and AgentPoirot maintained strong performance of 0.64 LLaMA-3-Eval compared to PandasAgent 0.58. This result demonstrates that the benchmark can scale to larger, more complex data volumes without degradation in the Agent’s performance.
>
> **4) Are there plans to incorporate real-world data into InsightBench?**
>
> - No, we do not plan to incorporate real-world data directly into InsightBench, as it may contain private or sensitive information. Instead, we prioritize the use of synthetic data to ensure privacy and security.
> - However, to ensure the benchmark's relevance to real-life scenarios, we will continue to use feedback from human experts from enterprises to vet and validate the synthetic data, ensuring it meets the distribution and the high-quality standards of real world data.
>
> (1/2) please see next comment for the rest of the rebuttal

---

> ### Author Response · Authors · 2024-11-20
> **Rebuttal (2/2) for Reviewer DA8a on InsightBench**
>
> **5) The LLaMA-3-Eval scoring method could benefit from additional explanation regarding its interpretability and stability, particularly in cases where scores may fluctuate with subtle prompt changes.**
>
> - LLaMA-3-Eval scores are highly interpretable as they are based on explicit criteria such as factuality, coherence and consistency. In terms of stability, our scores are based on 5 random runs where each leads to subtle prompt changes. The results showed very little variation, indicating consistent performance.
>
> - Regarding interpretability, here are some examples with justifications generated by the LLaMA-3 (added to the paper in Appendix C, Table 7):
>
>   - **Example 1 (for good generated insights):**
>      - **Ground Truth (GT):** There are significantly higher asset costs in the HR Department.
>      - **Generated:** The average cost of assets in the HR department is \\$4874.25, which is significantly higher compared to the average cost of \\$1967.26 in other departments.
>      - **LLaMA-3-Eval Score:** 0.81
>     - **Justification:** The generated text accurately conveys that the HR department has higher asset costs compared to other departments, with specific numerical values supporting the claim. The score is slightly less than 1 because the term "significant" is used in a general sense rather than in a statistical context. The generated text could be improved by providing additional context for the high asset costs.
>   - **Example 2 (for poorly generated insights):**
>       - **Ground Truth (GT):** There is a weak correlation between the number of users and the high cost of computer assets in the HR department.
>       - **Generated:** The distribution of asset types within the HR department includes 19 computers, 5 web servers, and 4 servers, with computers being the most prevalent asset type.
>       - **LLaMA-3-Eval Score:** 0.18
>       - **Justification:** The generated text does not address the correlation between the number of users and the cost of computer assets, which is the key point of the ground truth. Instead, it discusses the distribution of asset types, which is irrelevant to the claim, resulting in a low score.
>
> - This example highlights how LLaMA-3-Eval provides detailed and interpretable feedback, with scores reflecting both the relevance and accuracy of the generated insights.
>
> (2/2) End of Rebuttal

---

### Official Review · Reviewer_AkMH · 2024-11-05

**Soundness:** 3
**Presentation:** 3
**Contribution:** 3
**Rating:** 6
**Confidence:** 3

**Summary:**

This paper introduces InsightBench, a benchmark designed to evaluate the effectiveness of large language model-based (LLM-based) agents in performing comprehensive data analytics tasks. Unlike previous benchmarks, InsightBench focuses on end-to-end data analytics, assessing capabilities in question formulation, insight extraction, and summary generation across diverse business scenarios such as finance and incident management. InsightBench includes 100 datasets derived from the ServiceNow platform, enriched with specific insights and questions to evaluate an agent’s ability to recommend tasks and extract actionable insights. A baseline agent, AgentPoirot, is introduced and evaluated against other existing tools, with results showing it outperforms simpler query-based agents. The evaluation also incorporates LLaMA-3-Eval, an open-source alternative to GPT-4, offering a cost-effective and stable approach to assessing agent performance in text-based insight matching.

**Strengths:**

- InsightBench represents a significant step forward from single-query benchmarks, testing agents' holistic capabilities in an end-to-end analytics process more reflective of real-world data analysis.
- The paper showcases careful design in creating datasets with embedded trends, anomalies, and comprehensive goals that simulate actual enterprise challenges, making the benchmark realistic and versatile.
- By introducing LLaMA-3-Eval, the authors provide a stable and open-source alternative to GPT-4-based evaluations, reducing costs and ensuring reproducibility, a valuable contribution to the LLM community.
- AgentPoirot effectively demonstrates the viability of performing multi-step data analytics, including descriptive, diagnostic, predictive, and prescriptive tasks, positioning it as a useful baseline for future improvements in analytics agents.

**Weaknesses:**

- LLaMA-3-Eval offers advantages, but its consistency with GPT-4 evaluations could benefit from more rigorous comparison across various agent tasks beyond simple insight extraction.
- Despite the variety of themes, the benchmark may not fully capture the complexity of certain domains, potentially limiting generalizability to more complex or highly regulated fields like healthcare or law.
- Synthetic data generated from ServiceNow templates may lack the nuances of actual enterprise data, which could affect the evaluation's applicability to real-world use cases.

**Questions:**

- It would be valuable to test InsightBench on actual enterprise data and synthetic data to better understand its applicability to real-world tasks.
- Given the evolving nature of LLMs, a dynamic benchmark that updates or adjusts insight difficulty could be beneficial for testing agent adaptability over time.
- Further breakdown of evaluation scores (e.g., insight precision vs. general text similarity) could improve the interpretability and reliability of agent assessments.

**Details Of Ethics Concerns:**

The paper relies on synthetic data generated from ServiceNow templates, which mimics enterprise data structures. While synthetic data minimizes privacy risks, an ethics review could confirm whether the methods for data synthesis sufficiently anonymize or abstract the structures to prevent reverse engineering of sensitive data patterns.

---

> ### Author Response · Authors · 2024-11-20
> **Rebuttal for Reviewer AkMH on InsightBench**
>
> We appreciate the reviewer’s recognition of InsightBench’s end-to-end design as a real-world analytics benchmark, the realistic and versatile datasets it has which includes actual enterprise trends and anomalies, and the cost-effective, reproducible evaluation provided by LLaMA-3-Eval. We also thank the reviewer for acknowledging AgentPoirot’s potential as a foundation for multi-step data analytics agents.
>
> Below we address each of your concerns:
>
> **1) LLaMA-3-Eval offers advantages, but its consistency with GPT-4 evaluations could benefit from more rigorous comparison**
>
> - We have conducted that rigorous comparison in Table 5 in Appendix C, which displays score consistency between LLaMA-3-Eval and G-Eval across complex tasks that go beyond simple insight extraction such as detecting challenging trends and anomalies. Please let us know if there is any other specific comparison we could provide.
>
> **2) the benchmark may not fully capture the complexity of certain domains, potentially limiting generalizability to more complex or highly regulated fields like healthcare or law.**
>
> - Our benchmark is designed primarily for key business use cases, as stated in the title and abstract. However, our data generation pipeline and evaluation framework can be easily adapted for other complex domains like healthcare or law with the guidance of domain experts.
>
> **3) Synthetic data generated from ServiceNow templates may lack the nuances of actual enterprise data, which could affect the evaluation's applicability to real-world use cases.**
>
> - Our synthetic data was vetted by 12 ServiceNow experts that we partnered with that have deep knowledge of actual enterprise data to ensure it accurately represents real-world nuances.  Additionally, we have conducted a human evaluation study (detailed in Appendix B.4) where enterprise data specialists rated our synthetic data highly for its quality, coverage, and relevance.
>
> **4) It would be valuable to test InsightBench on actual enterprise data and synthetic data to better understand its applicability to real-world tasks.**
> -  We have found consistent relative performance results (w.r.t. L3-Eval metric) between AgentPoirot and PandasAgent on both our synthetic data and real ServiceNow datasets. The real-world data includes CSM, ITSM, and HR.
>
> |        Method    | Our Synthetic Data | Real World Data |
> |--------------------|---------------------------|------------------------|
> | PandasAgent |           0.54               |          0.58           |
> |  AgentPoirot   |           0.60               |         0.64            |
>
> - We unfortunately can’t include real enterprise data due to the proprietary nature of the data.
>
> **5) Given the evolving nature of LLMs, a dynamic benchmark that updates or adjusts insight difficulty could be beneficial for testing agent adaptability over time.**
> - Our benchmark includes datasets at varying difficulty levels: easy, medium, and hard. The best performance on the most difficult datasets remains low (at around 50%), which suggests that there is room for improvement from more advanced data analytics agents in the future.
>
> **6) Further breakdown of evaluation scores (e.g., insight precision vs. general text similarity) could improve the interpretability and reliability of agent assessments.**
> - To assess insight precision, we have included 15 datasets in our benchmark without dominant trends to evaluate whether an agent would produce false positives by generating insights where none exist. On these datasets, AgentPoirot scored an L3-Eval of 0.864, compared to PandasAgent’s 0.821, showing that AgentPoirot is better at avoiding false insights.
>
> - We also found that text similarity metrics (e.g., sentence-BERT embeddings with cosine distance) often yield high scores as they fail to measure factuality and accuracy of insights. Thus, we opted for the G-Eval metric, which has been shown to better align with human judgment for a more reliable evaluation [1].
>   - [1] Liu, Y., Iter, D., Xu, Y., Wang, S., Xu, R., & Zhu, C. (2023, December). G-Eval: NLG Evaluation using Gpt-4 with Better Human Alignment. In Proceedings of EMNLP.

---

### Meta-Review · Area_Chair_n9qL · 2024-12-21

**Metareview:**

This paper presents InsightBench, a benchmark that evaluates LLM-based agents in performing comprehensive data analytics. Overall the reviews are quite positive because of the paper's originality, clarity, and significance. There are also concerns on the synthetic data, dependency on structure, and scale, but the authors provided sufficient answers based on my judgement and the reviewer discussion afterwards. I therefore recommend accepting the paper.

**Additional Comments On Reviewer Discussion:**

The mains concerns were from reviewer nFSq, but the authors provided detailed responses.
- Synthetic data: the authors say the data was vetted by 12 ServiceNow experts and that they also conducted a human evaluation study by enterprise data specialists.
- Dependency on structure: the authors clarify that unstructured data like text/images/graphs is interesting, but beyond the scope of InsightBench.
- Scale: the authors explain the data generation pipeline is flexible and scalable, there are plans to open-source the benchmark, and there is already a partnership with ServiceNow.

These responses look reasonable to me.

---

### Decision · Program_Chairs · 2025-01-22

Accept (Poster)